# Non-Singularity of the Gradient Descent Map for Neural Networks with Piecewise Analytic Activations

**Alexandru Crăciun**
Technical University of Munich
`a.craciun@tum.de`

**Debarghya Ghoshdastidar**
Technical University of Munich
Munich Data Science Institute
Munich Center for Machine Learning
`ghoshdas@cit.tum.de`

## Abstract

The theory of training deep networks has become a central question of modern machine learning and has inspired many practical advancements. In particular, the gradient descent (GD) optimization algorithm has been extensively studied in recent years. A key assumption about GD has appeared in several recent works: the *GD map is non-singular* — it preserves sets of measure zero under preimages. Crucially, this assumption has been used to prove that GD avoids saddle points and maxima, and to establish the existence of a computable quantity that determines the convergence to global minima (both for GD and stochastic GD). However, the current literature either assumes the non-singularity of the GD map or imposes restrictive assumptions, such as Lipschitz smoothness of the loss (for example, Lipschitzness does not hold for deep ReLU networks with the cross-entropy loss) and restricts the analysis to GD with small step-sizes. In this paper, we investigate the neural network map as a function on the space of weights and biases. We also prove, for the first time, the non-singularity of the gradient descent (GD) map on the loss landscape of realistic neural network architectures (with fully connected, convolutional, or softmax attention layers) and piecewise analytic activations (which includes sigmoid, ReLU, leaky ReLU, etc.) for almost all step-sizes. Our work significantly extends the existing results on the convergence of GD and SGD by guaranteeing that they apply to practical neural network settings and has the potential to unlock further exploration of learning dynamics.

## 1 Introduction

Training deep neural networks involves optimizing highly non-convex loss functions over high-dimensional parameter spaces, a process that poses significant theoretical and practical challenges. Among these are the risks of gradient descent (GD) converging to saddle points or poor local minima, which could hinder the performance of trained models, especially when solutions associated with worst-case saddle points are significantly worse than those associated with worst-case local minima [Dauphin et al., 2014, Pascanu et al., 2014]. Recent theoretical advances have shed light on the dynamics of GD in such landscapes. One key result demonstrates that GD avoids converging to saddle points or maxima for almost all initializations, provided certain conditions are satisfied [Lee et al., 2019]. This supports empirical observations that GD tends to converge to local minima rather than get trapped in poor solutions, at least in the idealized setting in Lee et al. [2019]. Another line of research has explored the stability of minima, introducing Lyapunov exponent-like quantities that characterize whether GD converges to a given minimum from nearby initializations [Chemnitz and Engel, 2024]. These findings offer critical insights into why optimizing using GD often succeeds in practice despite the complexity and non-convex nature of the neural network loss.

39th Conference on Neural Information Processing Systems (NeurIPS 2025).

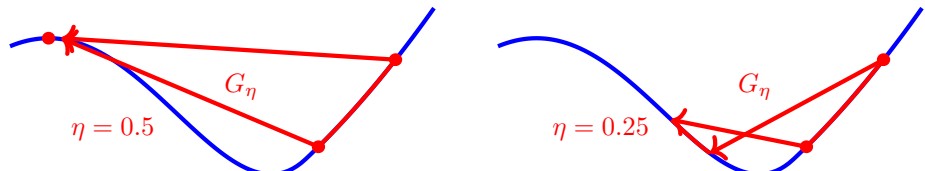

Figure 1: Illustrating the difference between singular and non-singular GD maps (different step-sizes). The loss used is $L(\theta) = 0.5\theta^4 - 3\theta^2 + 8$ if $-2 \leq \theta \leq 2$; $\theta^2$ otherwise. Left: for $\eta = 0.5$, the GD map is *singular*. The red interval gets mapped to a single point after one iteration. Right: for $\eta = 0.25$, the GD map is *non-singular*. The same interval is mapped not to a point, but to an interval.

Both of the analyses rely on the fact that the GD map — defined as $G_\eta(\theta) = \theta - \eta\nabla L(\theta)$, where $L$ is the loss function and $\eta > 0$ is the step-size — is non-singular. A map $G$ is *non-singular* if the preimage of any set of measure zero under $G$ also has measure zero; maps not having this property are called *singular*. Non-singularity of the GD map ensures that pathological behaviors (e.g., convergence to undesirable points) occur only on negligible sets. An example of the difference between the two concepts can be seen in Figure 1, which illustrates the effect of performing one optimization step. While non-singularity is relevant to many theoretical guarantees (such as the two works mentioned above), its validity in the context of neural networks has not been put on a solid footing. If the neural network uses an analytic activation function, it can be shown using only the standard tools of analysis that the gradient descent map is non-singular. However, if one employs a strictly piecewise analytic activation function like ReLU, the standard tools no longer work and a novel approach is needed.

In this paper, we prove that the (stochastic) GD map is non-singular for almost all step-sizes $\eta$ for neural networks using piecewise analytic activations and a number of different architectures, for example using fully connected, convolutional, or softmax attention layers, thereby validating the assumptions underpinning prior optimization theories in the context of practical neural network training. Applying the standard tools no longer works since the composition of two almost everywhere analytic functions need not be almost everywhere analytic itself (cf. Remark 5). However, we can make use of the layered nature of a neural network to prove that an analogue to the chain rule holds for the neural network function (Proposition 6). This result explains why the function is well-behaved for all parameter values except for a negligible set. With this insight, we extend the technique used to prove the non-singularity of the GD map in the case of neural networks with *strictly* analytic activations to networks using *piecewise* analytic activations (such as ReLU). Thus, our main result is:

**Theorem 1** (**Stochastic Gradient Descent Map for Neural Networks is Non-Singular**).
*Consider a deep neural network that consists of fully connected, convolutional, or attention layers and let the non-linear activations in the layers be piecewise analytic. Additionally, fix any data and any analytic loss function. Then, for almost all step-sizes $\eta$, any (S)GD map $G_\eta$ is non-singular.*

This contribution bridges a significant gap between theoretical optimization literature and practical neural network training. By establishing the non-singularity of the GD map, we extend the applicability of results on saddle-point and maxima avoidance [Lee et al., 2016, Panageas and Piliouras, 2016, Lee et al., 2019, Cheridito et al., 2022, Panageas et al., 2025] and on the stability of minima [Wu et al., 2018, Ma and Ying, 2021, Ahn et al., 2022, Chemnitz and Engel, 2024] to realistic deep learning settings, offering a rigorous explanation for the empirical success of GD. Of particular importance is the fact that the GD map is non-singular irrespective of the number of data points, hence theoretical results on the optimization properties of stochastic GD (SGD), such as those in Chemnitz and Engel [2024] can also be extended to realistic deep learning scenarios using our framework.

## 1.1 Related Works

Understanding the optimization dynamics of neural networks has been a central focus of machine learning research, blending classical mathematical tools with domain-specific insights. A well-known result in this area is Rademacher's Theorem [Rockafellar and Wets, 1998], which states that locally Lipschitz functions are differentiable almost everywhere. While this provides a general framework for studying smoothness, it does not directly address the unique structure of neural network losses, which are influenced by architectures and activation functions, nor does it provide more refined knowledge on the GD map (such as non-singularity), which is usually assumed in studying optimization.

Recent studies have focused on the piecewise nature of activation functions, such as ReLU, to better to understand how the layered structure of a deep neural network tessellates the input space into affine-linear regions as the parameters vary [Montufar et al., 2014, Balestriero et al., 2019, Balestriero and Baraniuk, 2020]. Software to efficiently visualize these regions has also been developed [Humayun et al., 2023]. These works have shed light on the geometric properties of the network function from the input space for fixed parameters: they estimate the number of affine linear regions of the resulting function and how this changes as the weights and biases of the network change. However, from an optimization point of view, one is interested in understanding the loss landscape. That is, for fixed data, one wants to know how the piecewise nature of the activation function splits the parameter space into different regions, separated by boundaries where the loss function is not differentiable.

In optimization theory, Panageas and Piliouras [2016], Lee et al. [2019] showed that first-order methods like GD almost always avoid saddle points under strict assumptions such as restricting the step-size to small values and requiring the loss function to be Lipschitz smooth. These are used to prove that the GD map is non-singular, which in turn is used to show that the set of initializations that converge to a saddle point has measure zero (if the set of saddle points has measure zero itself). Khromov and Singh [2023] have empirically investigated how the Lipschitzness of neural networks as functions from the input space changes during training and present bounds for a number of architectures and datasets. However, it is the Lipschitzness of the map from the parameter space which is needed to prove that GD avoids saddles and maxima. This condition is almost never true in neural networks (for example, it does not hold for deep ReLU networks with cross entropy as loss).

Ideas relating to a heuristic notion of stability of minima for GD and SGD have been explored in Wu et al. [2018], Ma and Ying [2021], where they conducted an investigation of how the batch-size and the step-size influence stability. Wu et al. [2018] train a model using GD until convergence and then switch to SGD, which, most of the time, converges to a different minimum (that generalizes better). Theoretical works such as Ahn et al. [2022] and Chemnitz and Engel [2024] start by assuming that the GD map is non-singular and use this to show that a notion of stability can be defined for minima. They conclude that stable minima attract nearby initializations, while unstable ones repel almost all starting points. The main contribution of Chemnitz and Engel [2024] is to extend the analysis of stability for GD to the stochastic setting, which is a non-trivial procedure. We mention that all the works referenced in this paragraph treat generic parametric models, not specializing to neural networks. Their assumption, that the gradient descent map is non-singular, while plausible, lacks rigorous justification in the neural network context, especially if the activation function is strictly piecewise analytic (such as ReLU, GELU, leaky ReLU, etc.). We also mention that stability has been empirically observed to be related to generalization ability [Hochreiter and Schmidhuber, 1997].

Closest to this paper is the work of Jentzen and Riekert [2022a,b, 2023]. While their work is more concerned with the convergence properties of gradient descent and gradient flow, they show that the loss function is continuously differentiable almost everywhere for deep neural networks using only ReLU as activation function under the assumption that the underlying function generating the data is polynomial. Our work expands on their results by allowing for any piecewise analytic activation function (including, but not limited to, sigmoid or ReLU), a broader class of network architectures (including convolutional and networks using attention layers), as well as a more general learning setting (we do not impose any restrictions on the data-generating process).

Our work builds on and extends these efforts by providing a formal proof that the gradient descent map is non-singular for neural networks with piecewise analytic activations, such as sigmoid, tanh, or ReLU. We show that the loss function is analytic almost everywhere and that the GD map preserves the measure zero property under preimages for almost all step-sizes. This result not only validates the assumptions in prior works but also establishes a stronger theoretical foundation for understanding optimization in deep learning, distinguishing our contribution from earlier heuristic analyses.

## 1.2 Organization of the Paper

In Section 2 we introduce the optimization task, different neural network architectures, and the (stochastic) gradient descent algorithm. In Section 3 we prove that the empirical loss function is almost everywhere analytic for neural networks with piecewise analytic activations. Next, in Section 4 we establish the non-singularity of the gradient descent map, and show why some values of the step-size should be avoided. Finally, in Section 5 we discuss the implications of this work, provide further illustrations and examples, and highlight directions for future research.

## 2 Setting

**Notation and Conventions.** Throughout this paper, we will only use the Lebesgue measure. Additionally, we use the abbreviation a.e. to mean almost everywhere. For a function $f : \mathbb{R}^m \to \mathbb{R}^n$ we denote by $Z(f) = \{x \in \mathbb{R}^m | f(x) = 0\}$ the set of points where $f$ is zero and if $U \subset \mathbb{R}^m$ is a set, we denote by $\partial U$ its topological boundary. For a finite set $I$, $|I|$ denotes the number of elements in $I$.

**Supervised Learning.** In the following, we consider a generic supervised learning problem. Let $F : \mathbb{R}^{n_\theta} \times \mathbb{R}^{n_0} \to \mathbb{R}^{n_D}$ be a *parametric* model, that is, we consider the first space in the Cartesian product of the domain of $F$ to be parameters. For each parameter $\theta \in \mathbb{R}^{n_\theta}$ we get a model that associates to an input $x \in \mathbb{R}^{n_0}$ an output vector $f_\theta(x) = F(\theta, x) \in \mathbb{R}^{n_D}$.

Given training data $((x_i, y_i))_{i=1}^m \subset \mathbb{R}^{n_0} \times \mathbb{R}^{n_D}$ and a loss function $l : \mathbb{R}^{n_D} \times \mathbb{R}^{n_D} \to \mathbb{R}$, the goal is to find a set of parameters $\theta^* \in \mathbb{R}^{n_\theta}$ that minimizes the empirical loss

$$L(\theta) = \frac{1}{m} \sum_{i=1}^m l(y_i, F(\theta, x_i)). \tag{1}$$

**Neural Networks.** Given a positive integer $D$ called the *depth* of the neural network, a sequence $N = (n_0, \ldots, n_D)$ of positive integers called the *widths*, and a sequence of piecewise analytic functions $\Sigma = \{\sigma_d : \mathbb{R} \to \mathbb{R}\}_{d=1}^D$ called the *activation functions*, we denote by $\mathcal{H}_{H,N,\Sigma}$ the hypothesis class of *feed-forward fully connected neural networks* (FF-FC) with a fixed depth and widths, and with piecewise analytic activation functions. These networks are the simplest and serve as a starting point for the more complicated architectures we will deal with.

We now describe how such a network works. An FF-FC network $f_\theta$ is given as

$$f_\theta(x) = \left( f^{(D)} \circ \cdots \circ f^{(1)} \right)(x).$$

Here $\theta = (W_1, b_1, \ldots, W_D, b_D)$ denotes the collection of parameters from each layer, where $b_i \in \mathbb{R}^{n_i}$ and $W_i \in \mathbb{R}^{n_i \times n_{i-1}}$ are called the *bias* and *weights* for that layer, respectively; we let $n_\theta = n_0 n_1 + \ldots + n_{D-1} n_D + n_1 + \ldots + n_D$ denote the total number of parameters.

The $d$-th network layer $f^{(d)}$ takes as input a vector $z^{(d-1)}$, the output of the previous network layer, and acts as follows:

$$f^{(d)}(z^{(d-1)}) = \sigma_d(W_d z^{(d-1)} + b_d),$$

where we apply the activation function $\sigma_d$ entry-wise.

In the rest of the paper, we will also be referring to convolutional or softmax attention layers. Such layers have been developed to solve specific tasks such as image recognition and natural language interpretation. A convolutional layer performs a convolution on the output from the previous layer using some kernel. Attention layers use a query, key, value system in order to find relevant tokens and better embeddings for sequential data. Deeper network architectures can be obtained by stacking together different types of layers. This setting is applicable to the transformer architecture [Vaswani et al., 2017] that interleaves convolutional and attention layers, followed by a fully connected network. Please refer to Appendix A for more details on different types of architectures and specialized layers.

**Bias is zero.** We mention here that for the rest of this paper, we set the bias terms in each layer equal to zero. This is done for clarity of exposition as dealing with the bias terms can be done in a straightforward way (cf. Remark 15 in Appendix B), but only makes the exposition harder to follow.

**Stochastic GD (SGD).** Gradient descent (GD) is defined as the map $G_\eta(\theta) = \theta - \eta \nabla L(\theta)$ for $\eta > 0$. For GD, we initialize the parameters $\theta_0$ according to a probability distribution on $\mathbb{R}^{n_\theta}$ and update them using the rule $\theta_{i+1} = G_\eta(\theta_i) = \theta_i - \eta \nabla L(\theta_i)$, where the loss is the one in (1). For stochastic GD (SGD), at each step $i$ we use only a sub-sample of $0 < n < m$ data points to compute the empirical loss. Hence, the new point is $\theta_{i+1} = G_{\eta,i}(\theta_i) = \theta_i - \eta \nabla L_i(\theta_i)$, where we randomly choose $i_1, \ldots, i_n \in [m], i_j \neq i_k$ if $j \neq k$, and the empirical loss computed on a mini batch is

$$L_i(\theta) = \frac{1}{n} \sum_{j=1}^n l(y_{i_j}, F(\theta, x_{i_j})).$$

# 3 Analyticity of the Loss Function

We now see that if the activation function of the network is analytic and the loss function is analytic as well, the empirical loss function is also analytic. This follows from the usual rules of calculus. It is no longer as obvious if that is still the case when the activation function is only piecewise analytic. The goal of this section is to show that the loss function for a neural network using piecewise analytic activations is itself analytic on a "big enough" set. To achieve this, we start by defining precisely what we mean by a piecewise analytic function and extend this definition from univariate to multivariate functions. Next, we highlight why a novel approach is needed and then show how we can leverage the layered structure of the neural network to derive analyticity guarantees. After the network function is properly understood, we finish by showing that the empirical loss function is analytic on a "big enough" set as well. Recall that a function $f : \mathbb{R}^m \supset U \to \mathbb{R}^n$ is analytic if for any point $x_0$ in its domain, it can be expressed as a power series converging to $f(x)$ for all $x$ in a neighborhood of $x_0$.

**Definition 2** (**Piecewise and almost eveywhere analytic functions**). We say a function $f : \mathbb{R} \to \mathbb{R}$ is *piecewise analytic* if there exists a strictly increasing sequence of real numbers $\{x_i\}_{i \in \mathbb{Z}}$ such that $f$ is analytic when restricted to any open interval $(x_i, x_{i+1})$. A function $f : \mathbb{R}^m \to \mathbb{R}^n$ is *analytic almost everywhere* if there exists an open set $U \subset \mathbb{R}^m$ such that $f|_U$ is analytic and the complement of $U$ has measure zero. By $S(f)$ we denote the points where $f$ is not analytic and $D(f) = \mathbb{R}^m \setminus S(f)$.

**Example 3** (**Some almost everywhere analytic functions**).

1. The sigmoid function $f : x \mapsto \frac{1}{1+\exp(-x)}$ is piecewise analytic with $D(f) = \mathbb{R}$.

2. The ReLU function $f : x \mapsto \max\{0, x\}$ is also piecewise analytic. Take $x_i = i$. We have that $f|_{(x_i, x_{i+1})}(x) = 0$ if $i < 0$ and $f|_{(x_i, x_{i+1})}(x) = x$ if $i \geq 0$. Here $S(f) = \{0\}$.

**Remark 4** (**Non-differentiable points**). In the previous definition, there may be multiple sets $\{U_i\}_{i \in I}$ on which $f$ is analytic and whose complements have measure zero. Denoting by $\mathcal{U}$ the collection of all such sets, we see that it is inductively ordered with respect to the usual inclusion of sets. An application of Zorn's lemma tells us that there exists a maximal such open set. It is not hard to check that it is also unique. This guarantees that the sets $D(f)$ and $S(f)$ are well-defined.

**Remark 5** (**Why a.e. analytic functions are problematic**). Observe that the chain rule does not hold for almost everywhere analytic functions. Whereas it is true that if $f$ is a.e. analytic and $g$ is analytic, we have that $g \circ f$ is a.e. analytic, it is not true that $f \circ g$ is a.e. analytic as the next (counter-)example shows. Let $h : \mathbb{R} \to \mathbb{R}$ be a nowhere differentiable function (e.g. the Weierstrass function) and define $f : \mathbb{R}^2 \to \mathbb{R}$ as $f(x, y) = h(x)$ if $y = 0$ and $f(x, y) = 0$ otherwise. Since $f$ is not analytic only on the abscissa ($y = 0$), it is a.e. analytic. Taking $g : \mathbb{R} \to \mathbb{R}^2$ to be the function $g(x) = (x, 0)$, we see that the composition $f \circ g$ is equal to $h$, hence it is not a.e. analytic.

As the next proposition shows, it turns out that we can use the structure of a neural network to derive an analogue to the chain rule for the function obtained by stacking together layers in a neural network. This is important since it will be used later to show that any gradient descent map obtained from a neural network loss function is non-singular.

**Proposition 6** (**Analogue of chain rule for neural networks**). *Let $D > 0$ be a positive number, $\{\sigma_i : \mathbb{R}^{n_i} \to \mathbb{R}^{n_i}\}_{i=1}^{D}$ be a collection of a.e. analytic maps and $\alpha \in \mathbb{R}^{d_0}$ a vector. Then the map defined recursively by*

$$f_D : \mathbb{R}^{n_1 \times n_0} \times \cdots \times \mathbb{R}^{n_H \times n_{D-1}} \to \mathbb{R}^{n_D}$$
$$(W_1, \ldots, W_D) \mapsto \sigma_D(W_D f_{D-1}(W_1, \ldots, W_{D-1})),$$

*with $f_1(W_1) = \sigma_1(W_1 \alpha)$ is analytic almost everywhere and the set $\partial Z(f_D)$ has measure zero.*

*Proof sketch.* We use induction. For the base case, one uses the fact that the map $m : W \mapsto W\alpha$ is a non-singular map (since it is a submersion). The points where $f_1$ is not differentiable have to lie in $m^{-1}(S(\sigma_1))$, hence they are of measure zero. A similar argument works for the boundary of $Z(f_1)$.

For the induction step, we split the domain of $f_{D-1}$: $B(f_{D-1}) = \partial Z(f_{D-1}) \cup S(f_{D-1})$, the "bad" points; $\text{int}(Z(f_{D-1}))$, the "nice" zeroes; and $N(f_{D-1}) = \text{dom}(f_{D-1}) \setminus (B(f_{D-1}) \cup \text{int}(Z(f_{D-1})))$, consisting of all "nice" non-zero points. The points in $B(f_{D-1})$ have measure zero, thus they can be neglected. One can then use the chain rule and properties of matrix multiplication to guarantee the analyticity of $f_D$ in a big enough region. A priori, this region might not be equal to $D(f_D)$, but it is big enough to guarantee that $S(f_D)$ has measure zero, showing that $f_D$ is a.e. analytic. $\qquad\square$

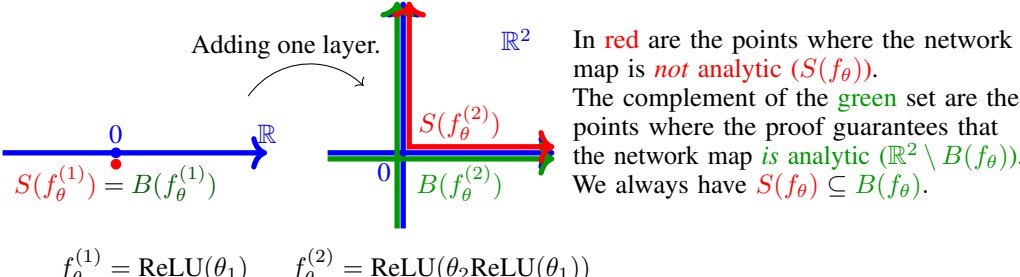

Adding one layer.

$\mathbb{R}^2$

$S(f_\theta^{(2)})$

In red are the points where the network map is *not* analytic ($S(f_\theta)$).
The complement of the green set are the points where the proof guarantees that the network map *is* analytic ($\mathbb{R}^2 \setminus B(f_\theta)$). We always have $S(f_\theta) \subseteq B(f_\theta)$.

$\mathbb{R}$

$0$

$S(f_\theta^{(1)}) = B(f_\theta^{(1)})$

$0$

$B(f_\theta^{(2)})$

$$f_\theta^{(1)} = \mathrm{ReLU}(\theta_1) \qquad f_\theta^{(2)} = \mathrm{ReLU}(\theta_2 \mathrm{ReLU}(\theta_1))$$

Figure 2: Illustration of the main idea in the proof of Proposition 6.

**Proposition 7 (Neural networks are a.e. analytic).** *Let $f_\theta : \mathbb{R}^{n_0} \to \mathbb{R}^{n_D}$ be an FF-FC, convolutional network or a network using softmax attention with piecewise analytic activation functions. Then for any input $x \in \mathbb{R}^{n_0}$, the map $\theta \mapsto f_\theta(x)$ is almost everywhere analytic.*

*Proof sketch.* The proof relies on Proposition 6. Let $f_\theta$ be the network map. Then we can directly apply Proposition 6 to show that the map $\theta \mapsto f_\theta(x)$ is a.e. analytic for any input $x \in \mathbb{R}^{n_0}$. We explain in Appendix B how that can be done for the convolutional and softmax attention layers. $\square$

The previous proposition is a key result, but it cannot be used directly to conclude that a GD map is non-singular. Theorem 1 will be proven in the next section. We now show that the empirical loss from a neural network is a.e. analytic. This is an important result since we need it to rove Theorem 1. We start with a lemma concerning the composition of a.e. analytic maps and analytic maps.

**Lemma 8 (Composition of a.e. analytic maps).** *Let $f : \mathbb{R}^m \to \mathbb{R}^n$ be an a.e. analytic function and $g : \mathbb{R}^n \to \mathbb{R}^l$ an analytic function. Then the composition $g \circ f : \mathbb{R}^m \to \mathbb{R}^l$ is a.e. analytic.*

*Proof.* We have that $f|_{D(f)}$ is analytic. Applying the chain rule, we have that $g \circ f|_{D(f)} = (g \circ f)|_{D(f)}$ is analytic, hence $g \circ f$ is analytic a.e. $\square$

**Corollary 9.** *Let $f_\theta : \mathbb{R}^{n_0} \to \mathbb{R}^{n_D}$ be any neural network with input dimension $n_0$, output dimension $n_D$, and $n_\theta$ parameters, using piecewise analytic activation functions. Then, given any dataset $\{(x_i, y_i)\}_{i=1}^m$ and any analytic loss function $l : \mathbb{R}^{n_D} \times \mathbb{R}^{n_D} \to \mathbb{R}$, the empirical loss given by*

$$L(\theta) = \frac{1}{m} \sum_{i=1}^n l(y_i, f_\theta(x_i))$$

*is almost everywhere analytic.*

*Proof.* The maps $\phi_i : \theta \mapsto (y_i, f_\theta(x_i))$ are a.e. analytic (since, by Proposition 7, each component is), thus we can use Lemma 8 to conclude that the compositions $l \circ \phi_i$ are a.e. analytic. The sum of a.e. analytic functions is a.e. analytic, thus it follows that $L$ is a.e. analytic. $\square$

## 4   (Stochastic) Gradient Descent Map is Non-Singular

We now use the results of the previous section to show that any gradient descent map is non-singular. Since this is true for any number of data points, we can immediately conclude that any SGD map $G_{\eta, i}$ is also non-singular. Because of this, we will state our results for GD, but keep in mind that they also hold for SGD as well. In particular, observe that we have an explicit description for any gradient descent map for almost all points. We start with a helpful lemma.

**Lemma 10 (Submersions are non-singular).** *Let $f : \mathbb{R}^n \supset U \to \mathbb{R}^n$ be a smooth map. If the set of critical points of $f$, i.e. points where the Jacobi determinant $\det(Df)$ is zero, has measure zero, then $f$ is non-singular.*

*Proof.* Suppose that $\det(Df(x)) \neq 0$ for any $x \in U$. This, together with the inverse function theorem [Lee, 2012, Theorem 4.5] tells us that $f$ is a local diffeomorphism.

Let $B \subset \mathbb{R}^n$ be any measure zero set. Because $f$ is a local diffeomorphism, around every point $x \in f^{-1}(B)$ there exists an open set $U_x \ni x$ such that $f$ restricted to it, $f|_{U_x} : U_x \to f(U_x)$, is a diffeomorphism. Since $f^{-1}(B) \subset \mathbb{R}^n$ and $\mathbb{R}^n$ is second countable, we can extract a countable subcover $\{U_{x_i}\}_{i=1}^{\infty}$ of $f^{-1}(B)$. Because $B$ has measure zero, so does each intersection $B_i = B \cap f(U_{x_i})$. Since $f$ restricted to $U_{x_i}$ is a diffeomorpishm, $f|_{U_i}^{-1}(B_i)$ has measure zero for all $i$. Observe that

$$f^{-1}(B) \subset \cup_{i=1}^{\infty} f|_{U_i}^{-1}(B_i).$$

A countable union of measure zero sets has measure zero and a subset of a measure zero set has measure zero, thus the special case is proved.

Going back to the general case, suppose that $V = \{\theta | \det(Df(\theta)) = 0\}$ has measure zero. The determinant is a continuous map and a singleton is closed, thus $V$ is closed. Then, the restriction of $f$ to the open set $U = \mathbb{R}^n \setminus V$ is a local diffeomorphism.

Let $B \subset \mathbb{R}^n$ be any measure zero set. Its preimage under $f$ can be written as

$$f^{-1}(B) = f|_V^{-1}(B) \cup f|_U^{-1}(B).$$

The first term in the union is the intersection of a set with $V$, which has measure zero by assumption, thus it has measure zero. The second term has measure zero using the special case proved above. It follows that $f^{-1}(B)$ has measure zero. $\qquad \square$

We use the previous lemma to show that *the* gradient descent map for analytic loss functions is non-singular. We emphasize that there is a *unique* gradient descent map if the loss is analytic.

**Proposition 11 (GD map for analytic loss functions is non-singular for almost all step-sizes).** *Let $L : \mathbb{R}^{n_\theta} \supset U \to \mathbb{R}$ be an analytic function and assume without loss of generality that $U$ is connected. Then, for almost all $\eta > 0$, the GD map $G_\eta : x \to x - \eta \nabla L(x)$ is non-singular.*

*Proof.* If $U$ has more than one connected component, we will show that $G_\eta|_{U_i}$, where $U_i$ is a connected component of $U$, is non-singular. Since $U$ has at most countably many connected components, it follows that $G_\eta$ is non-singular as well.

If all the eigenvalues $\lambda_1, \dots, \lambda_{n_\theta}$ of the $H_L$, the Hessian of $L$, are constant, then for any $\eta \notin \{1/\lambda_1, \dots, 1/\lambda_{n_\theta}\}$ we have that $\det(I - \eta H_L) = \prod_{i=1}^{n_\theta}(1 - \eta \lambda_i)$ is non-zero. This means that the Jacobi determinant $\det DG_\eta = \det(I - \eta H_L)$ is always non-zero, hence $G_\eta$ is non-singular.

If at least one eigenvalue $\lambda$ is not constant, there exist two points $\theta_1, \theta_2 \in U$ such that $\lambda(\theta_1) \neq \lambda(\theta_2)$. Take any analytic path $\gamma \colon \mathbb{R} \to U$ with $t_1 \neq t_2 \in \mathbb{R}$, $\gamma(t_1) = \theta_1$ and $\gamma(t_2) = \theta_2$. This can be done since we can interpolate any finite number of points using analytic functions. Since $\gamma$ and the map $\theta \mapsto H_L(\theta)$ are both analytic, their composition, $t \mapsto H_L \circ \gamma(t) \in \mathbb{R}^{n_\theta \times n_\theta}$, is analytic as well. In this case, the eigenvalues of $H_L \circ \gamma$ can be globally parameterized by analytic functions, i.e., there exist analytic functions $\lambda_i \circ \gamma \colon \mathbb{R} \to \mathbb{R}$ equal to the eigenvalues of $H_L \circ \gamma(t)$ for all $t$ [Katō, 1995].

For a non-constant analytic function, the set where it is equal to a constant $c \in \mathbb{R}$, $Z(\lambda_i \circ \gamma - c) \subset \mathbb{R}$ has measure zero. For any $\bar{t} \in \mathbb{R} \setminus \cup_{i=1}^{n_\theta} Z(\lambda_i \circ \gamma) \subset \mathbb{R}$, it holds that $\lambda_i \circ \gamma(\bar{t}) \neq c$. In particular, for any $\eta > 0$ there exists $\bar{\theta} = \gamma(\bar{t}) \in \mathbb{R}^{n_\theta}$ such that $\det(I - \eta H_L(\bar{\theta})) \neq 0$. We have showed that the analytic map $\theta \mapsto \det(I - \eta H_L(\theta))$ is not constant, thus its set of zeros has measure zero. We conclude that $G_\eta$ is invertible almost everywhere, i.e., the set of points where the Jacobi determinant is zero has measure zero, whence we can apply the previous lemma to show that $G_\eta$ is non-singular. $\quad \square$

**Corollary 12 (GD map is non-singular).** *Let $L : \mathbb{R}^{n_\theta} \to \mathbb{R}$ be an a.e. analytic function, $\eta > 0$, and $G_\eta : \mathbb{R}^{n_\theta} \to \mathbb{R}^{n_\theta}$ be a function such that for all $\theta \in D(L)$ we have that $G_\eta(\theta) = \theta - \eta \nabla L(\theta)$. Then, for almost all $\eta > 0$, any gradient descent map $G_\eta$ is non-singular.*

*Proof.* By the previous proposition, $G_\eta$ is non-singular when restricted to $D(L)$ for almost all values of $\eta$. If $B$ is any measure zero set in $\mathbb{R}^{n_\theta}$, we have that $G_\eta^{-1}(B) = G_\eta|_{D(L)}^{-1}(B) \cup G_\eta|_{S(L)}^{-1}(B)$. The first set in the union has measure zero since $G_\eta$ is non-singular on $D(L)$ and the second one as well since $S(L)$ has measure zero. $\qquad \square$

# 5 Discussion

In this paper, we have demonstrated that for neural networks with piecewise analytic activation functions, the GD map $G_\eta(\theta) = \theta - \eta\nabla L(\theta)$ and the SGD maps $G_{\eta,i}$ are non-singular for almost all step-sizes $\eta$. This result has profound implications for the theoretical understanding of neural network optimization. By proving that the loss function is analytic almost everywhere and that the (S)GD map preserves sets of measure zero under preimages, we confirm a key assumption in prior works, such as those by Lee et al. [2019] and Chemnitz and Engel [2024]. This bridges a critical gap between theoretical guarantees and the practical success of GD in training deep networks.

Our findings imply that, for almost all initializations and step-sizes, the optimization trajectory avoids pathological behaviors — such as convergence to saddle points or unstable minima[1] — aligning with empirical observations of (S)GD's effectiveness. The proof relies on two key insights: first, the piecewise analyticity of the activation functions ensures the loss is analytic outside a set of measure zero; second, the analytic properties of the loss in the "nice" regions ensure the GD map is invertible almost everywhere, except for a negligible number of values for the step-size. While our result is robust across almost all step-sizes, we note that non-singularity may fail at specific values for the step-size (where $\eta = 1/\lambda_i$ for an eigenvalue $\lambda_i$ of the Hessian of $L$), which are negligible in practice.

To illustrate the connection between our result and the stability of minima, we first recall a key result from Chemnitz and Engel [2024]. They show that for a generic parametric model, if the GD map is non-singular and the set of global minima $M$ forms a manifold, then the stability of a minimum $\theta \in M$ is completely determined by a computable quantity. Our Theorem 1 provides the missing piece to rigorously apply this framework to neural networks. By establishing the non-singularity of the (S)GD map for networks with piecewise analytic activations, we can now state the following:

**Corollary 13.** *Let the assumptions be as in Theorem 1. Additionally, suppose the data is generic and let $M$ be the set of global minima of the loss function $L$. Then, for almost all step-sizes, there exist functions $\mu(\theta)$ and $\lambda(\theta)$ such that a global minimum $\theta \in M$ is stable under GD or SGD only if $\mu(\theta) < 0$ or $\lambda(\theta) < 0$, respectively. Conversely, if $\mu(\theta) > 0$ or $\lambda(\theta) > 0$, then $\theta$ is unstable.*

The definitions of $\mu$ and $\lambda$ are the same as those in Chemnitz and Engel [2024]. For a discussion of "generic data" see Cooper [2021]; note that it is reasonable to assume that noisy data is generic.

We now use this corollary to analyze the training dynamics in a few examples. We first compute the functions $\mu$ and $\lambda$ for the simple setting with loss function $(\theta_1, \theta_2) \mapsto 3.51(1-\text{ReLU}(\theta_2\text{ReLU}(\theta_1)))^2$ (corresponding to a two-layer ReLU neural network as described in Appendix C). We then use these functions to: 1) determine the stability of periodic trajectories under GD as the step-size $\eta$ changes (Figure 3), and 2) show that for a fixed step-size, a global minimum can be stable under GD but unstable under SGD, and vice-versa (Figure 4). The next two paragraphs elaborate on this.

**Application: periodic trajectories.** The composition of non-singular maps is non-singular as well, hence the stability analysis of minima can be extended to periodic trajectories. The existence of such trajectories can be motivated twofold: (i) the use of large step-sizes (as incentivized by results from Li et al. [2019], Mohtashami et al. [2023]) leads to minima being stable only if they are flat (cf. Chemnitz and Engel [2024]); (ii) it has been empirically observed that the eigenvalues of the Hessian matrix of the loss oscillate even after numerous training epochs [Cohen et al., 2021, 2023], which means that the trajectory is not converging. Our results provide a comprehensive framework for dealing with such cases. We illustrate this through an experiment described in Appendix C. We assume a 2-layer ReLU network $f_\theta(x) = \text{ReLU}(\theta_2\text{ReLU}(\theta_1 x))$, a quadratic loss, and data sampled from a linear function. For $k > 0$, we are interested in $k$-periodic points, that is points $\theta^* \in \mathbb{R}^2$ such that $G_\eta^k(\theta^*) = \theta^*$. The sequence of iterates $\theta^*, G_\eta(\theta^*), \ldots, G_\eta^{k-1}(\theta^*)$ is then a periodic trajectory. A periodic trajectory is stable if there is a region $U$ around $\theta^*$ such that initializing in $U$ leads to trajectories that stay close to the periodic one. In Figure 3 on the right we plot trajectories with the same initialisation, converging to different periodic trajectories for different step-sizes. In Figure 3 on the left we plot the bifurcation diagram for periodic trajectories lying on the diagonal ($\theta_1 = \theta_2$). As the step-size increases, new periodic trajectories are created and the old ones become unstable.

---

[1]One also has to show that the saddle points and the unstable minima lie in a measure zero set. When $n_\theta \leq m$ this is true since such points are isolated. In the overparameterized case, when $n_\theta > m$, proving this requires more care. Cooper [2020, 2021] have proven this for neural networks with smooth activations.

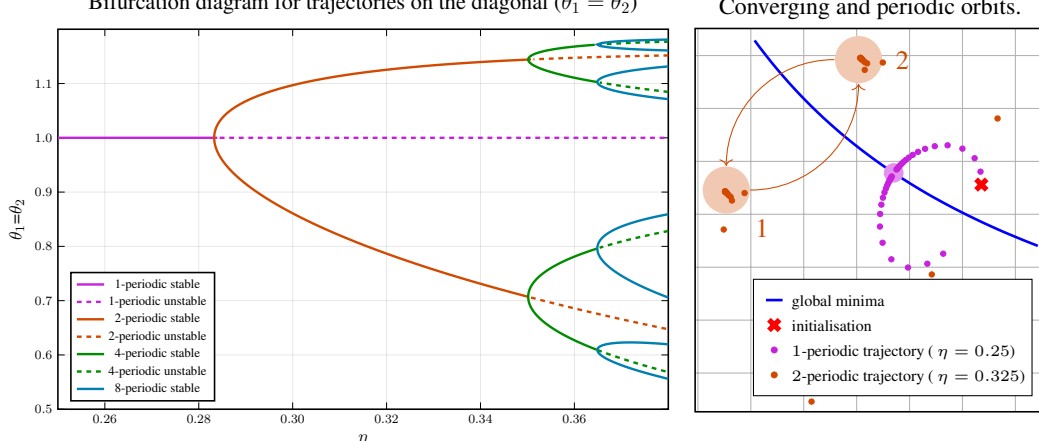

Figure 3: Left: periodic trajectories for GD on $L(\theta_1, \theta_2) = 3.53(1 - \text{ReLU}(\theta_2 \text{ReLU}(\theta_1)))^2$. As the step-size $\eta$ increases, higher order orbits appear and the lower order ones become unstable. Right: for the same initialisation, but different $\eta$, two trajectories, one that converges and one that oscillates.

**Stochastic and adaptive variants of GD.** Our theoretical framework robustly covers SGD and settings where a learning rate schedule is used. Theorem 1 establishes the non-singularity of the GD map $G_{\eta,i}$ for *any fixed data and any batch size.* An SGD step is mathematically equivalent to a full GD step performed on a loss function computed over a mini-batch of data. Since our result holds *irrespective* of the number of data points used to compute the loss, it directly applies to each step of an SGD trajectory, where the mini-batch changes at each iteration. The map for each step is therefore non-singular (for almost all $\eta$), and the composition of these maps, which describes the full SGD trajectory, is also non-singular. An optimisation process using GD over $T$ steps with a learning rate schedule $(\eta_1, \eta_2, \ldots, \eta_T)$ can be described by the composite map $G = G_{\eta_T} \circ \cdots \circ G_{\eta_1}$. This is a composition of non-singular maps, hence their composition is also guaranteed to be non-singular. Therefore, our analysis provides a rigorous foundation for understanding optimization dynamics not just for a fixed step-size, but for any practical scheme that relies on composing gradient descent steps.

**Application: stable minima of GD vs SGD.** Results from Wu et al. [2018] have shown that GD and SGD can have different stable global minima. For the setting described above, we can analytically compute the quantities introduced in Chemnitz and Engel [2024] which determine stability (see Appendix C). The relation between the minima of GD and SGD can be quite complex, so we illustrate two interesting cases. In Figure 4 on the left, we see that there are fewer stable global minima for SGD than for GD. In Figure 4 on the right, we see a setting where the stable minima for GD and SGD do not overlap at all. It remains an open problem to determine if the stable minima for GD and SGD can be related in a general setting, or if such a relation has to be determined on a case-by-case basis.

**Different architectures.** While our results already hold for a diverse number of layers (fully connected, convolutional or using softmax attention) this does not cover all cases used in practical settings. An in depth investigation of other architectures may reveal additional structural properties that make it possible to prove that the neural network map is analytic a.e. for those architectures as well. We think that an extension to graph and residual neural networks might consist only of routine applications of the ideas already present in our proofs. However, for recurrent neural networks, we see that our techniques are not enough and another approach is needed. One might approach this setting by viewing the recurrent network $f : \mathbb{R}^{n_0} \to \mathbb{R}^{n_D}$ as a larger fully connected network $F : \mathbb{R}^N \to \mathbb{R}^{n_D}$ ($N > n_0$) for which some weights are forced to be equal (cf. Appendix A). Proposition 6 guarantees that $S(F)$, the singularities of $F$ have measure zero in $\mathbb{R}^N$. We have that $S(f) \subseteq \mathbb{R}^{n_0} \cap S(F)$ (where $\mathbb{R}^{n_0}$ is viewed as a subset of $\mathbb{R}^N$). Hence, if $\mathbb{R}^{n_0} \cap S(F)$ has measure zero, so must $S(f)$ too. However, it is not clear that this intersection is negligible and a more in depth investigation of $S(F)$ is needed to be sure. Even with this current limitation of our proof techniques, we conjecture that the GD map is non-singular for the loss landscape of a recurrent neural network.

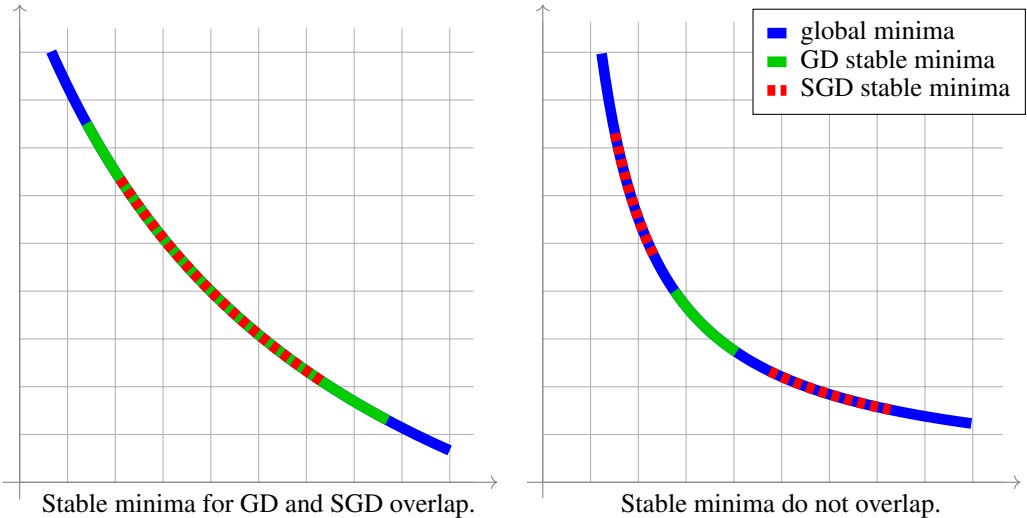

| Stable minima for GD and SGD overlap. | Stable minima do not overlap. |

Figure 4: Illustrating the different stable minima for GD and SGD. In blue are the global minima for $L(\theta_1, \theta_2) = 3.53(1 - \mathrm{ReLU}(\theta_2 \mathrm{ReLU}(\theta_1)))^2$; in green are the minima stable for GD; in red those stable for SGD. Left: the stable minima for SGD are a proper subset of the stable minima for GD. Right: the stable minima for SGD and GD do not intersect. This change is due to different step-sizes ( $\eta_{\text{left}} = 0.15$ vs $\eta_{\text{right}} = 0.3$) and different probabilities for generating the data (cf. Appendix C).

**Other optimization algorithms.** The techniques developed in this paper could be applied to other optimization methods such as mirror descent or proximal point algorithms, potentially broadening the scope of non-singularity guarantees. These alternative optimization methods share with GD the core principle of iteratively updating a solution to minimize a loss function (commonly using only first-order information like the gradient). Lee et al. [2019] have analysed the convergence properties of such algorithms, albeit in the same restricted setting they have for GD (i.e. requiring the loss function to be Lipschitz smooth). To extend our results to different optimization algorithms, one has to modify the proofs in Section 4 of our work. In particular, it suffices to investigate the case when the empirical loss function $L$ is analytic, since once it is proven that the optimizer map is non-singular for such losses, the same proof as that for Corollary 12 can be used to extend it to a.e. analytic losses.

## Acknowledgements

This work has been partially supported by the German Research Foundation (DFG) through the Priority Program SPP 2298 (project GH 257/2-2). We also thank the reviewers for their feedback.

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

## A  Different Architectures

In this short appendix, we expand on the discussion from Section 2 regarding the different architectures besides the ones made up only of fully connected layers.

**Convolutional Networks.** There exist also other kinds of neural network architectures, more specialized to certain tasks. The first which we introduce are *convolutional* neural networks Le Cun et al. [1989]. This kind of networks are different from FF-FC networks because they have convolutional layers. A detailed description of such networks can be found in Goodfellow et al. [2016], but for our purposes it suffices to know that a convolutional layer is such that some entries in the weight matrix are equal to each other. For example, for a univariate convolution, each row of the weight matrix is constrained to be equal to the row above shifted by one element Goodfellow et al. [2016].

**Recurrent Networks.** To better learn time-series, *recurrent* neural networks Rumelhart et al. [1986] have been introduced. The idea behind them is that if the input to the network is a series $x = (x_1, \ldots, x_T)$, the network might benefit by using the intermediate activations it has computed at previous steps when computing the current activation. More explicitly, if we denote by $f_t^{(d)}(x_1, \ldots, x_t)$ the output of the $d$-th layer of the network at time $t$, then the output of the $(d+1)$-th layer of the network at time $t + 1$ is defined recursively by the following relation

$$f_{t+1}^{(d+1)}(x_1, \ldots, x_{t+1}) = \sigma(W_{d+1} f_{t+1}^{(d)}(x_1, \ldots, x_{t+1}) + U_{t+1} f_t^{(d+1)}(x_1, \ldots, x_t) + b_{d+1}),$$

where $\sigma$ is applied entry-wise and the base case is $f_1^{(1)}(x_1) = \sigma(W_1 f_0^{(0)} + U_1 x_1 + b_1)$, with $f_0^{(0)}$ a constant. The new matrices we introduced $U_i \in \mathbb{R}^{n_i} \times \mathbb{R}^{n_i}$ represent the recurrent connections.

In this case, the outputs of the network at any time-step can be used when computing the empirical loss function. That is, for every input $x_i$ (we index different inputs with the subscript $i$), we have a set $I_i \subset [T]$ of time instances, and we can write the empirical loss as

$$L(\theta) = \frac{1}{m} \sum_{i=1}^{m} \frac{1}{|I_i|} \sum_{t \in I_i} l(y_i, f_t^{(D)}(x_{i,1}, \ldots, x_{i,t})).$$

**Networks using Attention.** As an alternative to recurrent networks, *attention* layers have been developed [Vaswani et al., 2017]. Such a layer has three learnable matrices $W_q, W_k$, and $W_v$, and works as follows: given a sequence $x_1, \ldots, x_n \in \mathbb{R}_0^n$, it outputs a sequence $y_1, \ldots, y_n \in \mathbb{R}_D^n$. Denoting by $X$ the input matrix, i.e. the matrix having the $x_i$'s as columns, we obtain three matrices: $Q = W_q X, K = W_k X, V = W_v X$ called the queries, keys, and values, respectively. The output is:

$$Attention(Q, K, V) = softmax(\frac{QK^T}{\sqrt{n}})V,$$

where the $softmax$ function is independently applied to every row of its argument. The columns of this matrix are the output sequence $y_1, \ldots, y_n$. After applying a number of attention layers, we may use an FF-FC network at the end. This is usually done by concatenating the output sequence into a vector $y = y_1 * \ldots * y_n$ and using this as the input to an FF-FC network. The important result for our analysis is to observe that for any input sequence $x_1, \ldots, x_n$, the map determined by an attention layer on the parameters $(W_q, W_k, W_v) \mapsto Attention(Q, K, V)$ is *analytic*.

## B  Proofs of Main Results

In this appendix we provide the full proofs of Propositions 6 and 7. There is one difference from the result stated in the main body of the text. Instead of proving the result for a.e. analyticity, we prove a more general result, showing that the smoothness properties of the activation function carry over to the neural network map almost everywhere. What we mean by this is that if the activation function is piecewise $\mathcal{C}^k$, where $k$ can be a positive integer (denoting $k$-times differentiable functions), $\infty$ (denoting smooth functions), or $\omega$ (denoting analytic functions), then the neural network function is also almost everywhere $\mathcal{C}^k$. The definition of an almost everywhere $\mathcal{C}^k$ function is analogous to Definition 2. We begin by restating Proposition 6.

**Proposition 14 (Analogue of chain rule for fully connected layers).** *Let $D > 0$ be a positive number and $\{\sigma_i : \mathbb{R}^{n_i} \to \mathbb{R}^{n_i}\}_{i=1}^D$ be a collection of a.e. $\mathcal{C}^k$ maps with the property that $\partial Z(\sigma_i)$ has*

*measure zero and $\alpha \in \mathbb{R}^{n_0}$ a vector. Then the map $f_D$ defined recursively by*

$$f_d : \mathbb{R}^{n_1 \times n_0} \times \cdots \times \mathbb{R}^{n_d \times n_{d-1}} \to \mathbb{R}^{n_d}$$
$$(W_1, \ldots, W_d) \mapsto \sigma_d(W_d f_{d-1}(W_1, \ldots, W_{d-1})),$$

*with $f_1(W_1) = \sigma_1(W_1 \alpha)$ is $\mathcal{C}^k$ almost everywhere and the set $\partial Z(f_D)$ has Lebesgue measure zero.*

*Proof.* We will prove this proposition by induction on $D$. If $D = 1$, we have that $f_1(W_1) = \sigma_1(W_1 \alpha)$.

If $\alpha = 0$, then $f_1(W_1) = \sigma_1(0)$ is a constant function, hence it is $\mathcal{C}^k$. Also, we have that $Z(f_1)$ is either the whole domain of $f_1$ or the empty set, hence $\partial Z(f_1)$ has measure zero.

In the case $\alpha \neq 0$, the set of points $S(f_1) = \{W_1 \alpha \in S(\sigma_1)\}$ (*a priori* this set need not be equal to the set of points where $f_1$ is *not* $\mathcal{C}^k$; we show in this paragraph that they *do* coincide) has measure zero and is also closed. This follows since the multiplication map $M_1 : W_1 \mapsto W_1 \alpha$ is non-singular. With this definition, we have that $f_1$ is the composition $f_1 = \sigma_1 \circ M_1$. Taking any $W_1 \notin S(f_1)$, we have that $M_1$ is $\mathcal{C}^k$ in a neighborhood of $W_1$ and that $\sigma_1$ is $\mathcal{C}^k$ in a neighborhood of $W_1 \alpha = M_1(W_1)$, hence we can apply the chain rule to conclude that $f_1$ is $\mathcal{C}^k$ in a neighborhood of $W_1$. We see that $f_1$ is $\mathcal{C}^k$ in the complement of $S(f_1)$. But $S(f_1)$ is a closed set and has measure zero, hence $f_1$ is $\mathcal{C}^k$ almost everywhere.

We now show that $\partial Z(f_1)$ has measure zero. Looking at the set $Z(f_1) = \{W_1 | W_1 \alpha \in Z(\sigma_1)\}$, we see that $Z(f_1) = M_1^{-1}(Z(\sigma_1))$. We also have that $W \in \partial Z(f_1)$ implies that $M_1(W) \in \partial Z(\sigma_1)$. Hence, we see that $\partial Z(f_1) \subset M_1^{-1}(\partial Z(\sigma_1))$. Since $M_1$ is a non-singular map (it is a submersion), it follows that $\partial Z(f_1)$ has measure zero.

Now we proceed to the induction step. Assume that $f_{D-1}$ is $\mathcal{C}^k$ a.e. and that $\partial Z(f_{D-1})$ has measure zero. Define the following sets:

1. The set of "bad" points of $f_{D-1}$:

$$B(f_{D-1}) = \partial Z(f_{D-1}) \cup S(f_{D-1}).$$

   By assumption, we have that $B(f_{D-1})$ is a measure zero set.

2. The set of "nice" points of $f_{D-1}$ which are not roots:

$$N(f_{D-1}) = \mathrm{dom}(f_{D-1}) \setminus (B(f_{D-1}) \cup \mathrm{int}(Z(f_{D-1}))),$$

   where $\mathrm{dom}(f_{D-1})$ is the domain of $f_{D-1}$.

We also use the following notation

$$x_D = (W_1, \ldots, W_D) \text{ and } y_{D-1} = W_D f_{D-1}(W_1, \ldots, W_{D-1}).$$

We now consider two cases:

1. If $x_{D-1} \in N(f_{D-1})$, then take any $W_D$ such that $W_D f_{D-1}(x_{D-1}) \notin S(\sigma_D)$. Such a $W_D$ exists since $f_{D-1}(x_{D-1})$ is not zero. In fact, the complement of the set of all such points has measure zero since it is the preimage of the set $S(\sigma_D)$ under the multiplication map $M_D : W_D \mapsto W_D f_{D-1}(x_{D-1})$, which is a non-singular map.

   We will write $f_D$ as a composition of maps and then show that we can use the chain rule to conclude that $f_D$ is $\mathcal{C}^k$ in a neighborhood of $x_D$. Thus, consider the following maps:

$$f_{D-1} \times \mathrm{id} : \mathbb{R}^{n_1 \times n_0} \times \ldots \times \mathbb{R}^{n_D \times n_{D-1}} \to \mathbb{R}^{n_{D-1}} \times \mathbb{R}^{n_D \times n_{D-1}}$$
$$(W_1, \ldots, W_D) \mapsto (f_{D-1}(W_1, \ldots, W_{D-1}), W_D)$$

$$M_D : \mathbb{R}^{n_{D-1}} \times \mathbb{R}^{n_D \times n_{D-1}} \to \mathbb{R}^{n_D}$$
$$(x_{D-1}, W_D) \mapsto W_D x_{D-1}.$$

   We see that $f_D$ can be written as the composition $\sigma_D \circ M_D \circ (f_{D-1} \times \mathrm{id})$. By assumption, we have that $f_{D-1} \times \mathrm{id}$ is $\mathcal{C}^k$ in a neighborhood of $x_D$; $M_D$ is $\mathcal{C}^k$ in its whole domain, in

particular in an open neighborhood of $(f_{D-1}(x_{D-1}), W_D)$; and also $\sigma_D$ is $\mathcal{C}^k$ on an open neighborhood of $y_D$ by assumption. Hence, we can apply the chain rule to conclude that $f_D$ is $\mathcal{C}^k$ in an open neighborhood of $x_D$.

2. If $x_{D-1} \in \text{int}(Z(f_{D-1}))$, then we can use a different approach. Since it is an interior point, there exists an open neighborhood $U \ni x_{D-1}$ such that $U \subset \text{int}(Z(f_{D-1}))$. We immediately see that on this neighborhood $f_D$ is constant, hence it is $\mathcal{C}^k$.

We can now describe the set of points where the previous two cases guarantee that $f_D$ is differentiable. Start by defining $\Delta$ to be the set

$$\Delta = \{(x_{D-1}, W_D) \in N(f_{D-1}) \times \mathbb{R}^{n_D \times n_{D-1}} | W_D f_{D-1}(x_{D-1}) \in S(\sigma_D)\}.$$

It is closed since $S(\sigma_D)$ is closed and the map $M_D : (x_{D-1}, W_D) \mapsto W_D f_{D-1}(x_{D-1})$ is continuous. Since this map is measurable and so is $S(\sigma_D)$, it follows that $\Delta$ is also measurable. In the first case above, we proved that for any $x_{D-1} \in N(f_{D-1})$ the intersection $\Delta \cap \{x_{D-1}\} \times \mathbb{R}^{n_D \times n_{D-1}}$ has measure zero. Use can use a special case of the Fubini's theorem to conclude that $\Delta$ has measure zero as well.

In case 1 we prove that $f_D$ is $\mathcal{C}^k$ on $(N(f_{D-1}) \times \mathbb{R}^{n_D \times n_{D-1}}) \setminus \Delta$. In case 2, we showed that $f_D$ is $\mathcal{C}^k$ on $\text{int}(Z(f_{D-1})) \times \mathbb{R}^{n_D \times n_{D-1}}$. Putting everything together, we see that $F$ is $\mathcal{C}^k$ on an open set whose complement has measure zero, namely $\Delta \cup B(f_{D-1}) \times \mathbb{R}^{n_D \times n_{D-1}}$. This shows that $f_D$ is $\mathcal{C}^k$ almost everywhere.

We now prove the second part, namely that $\partial Z(f_D)$ has measure zero. We have that $Z(f_D) = \{(x_{D-1}, W_D) | W_D f_{D-1}(x_{D-1}) \in Z(\sigma_D)\}$. Start by observing that if $x_D \in \partial Z(f_D)$, then $y_D \in \partial Z(\sigma_D)$. This follows since $M_D$ is a continuous map. This means that $\partial Z(f_D) \subset M_D^{-1}(\partial Z(\sigma_D))$. We consider three cases:

1. If $x_{D-1} \in N(f_{D-1})$, we can look at the derivative of $M_D$ at a point $(x_{D-1}, W_D) \in N(f_{D-1}) \times \mathbb{R}^{n_D \times n_{D-1}}$. For any $(\bar{x}_{D-1}, \bar{W}_D)$ in the tangent space at $x_D$, we have:

$$d_{x_D} M(\bar{x}_{D-1}, \bar{W}_D) = \bar{W}_D f_{D-1}(x_{D-1}) + W_D d_{x_{D-1}} f_{D-1}(\bar{x}_{D-1}).$$

It is surjective since $f_{D-1}(x_{D-1}) \neq 0$. Hence, $M$ is non-singular on $N(f_{D-1}) \times \mathbb{R}^{n_D \times n_{D-1}}$. Since $\partial Z(\sigma_D)$ has measure zero, we have that $N(F) \times \mathbb{R}^{n_D \times n_{D-1}} \cap \partial Z(f_D)$ also has measure zero.

2. If $x_{D-1} \in \text{int}(Z(f_{D-1}))$, then $W_D f_{D-1}(x_{D-1})$ is constant for any $W_D$. Thus, either all points $y_{D-1}$ are roots of $\sigma_D$ or none. In both cases the set $\text{int}(Z(f_{D-1})) \times \mathbb{R}^{n_D \times n_{D-1}} \cap \partial Z(f_D)$ has measure zero.

3. If $x_{D-1} \in B(f_{D-1})$, then the set $B(f_{D-1}) \times \mathbb{R}^{n_D \times n_{D-1}}$ has measure zero, hence its intersection with $\partial Z(f_D)$ also has measure zero.

Since the domain of $f_D$ is a disjoint union of the three sets we have considered in the previous cases and since the intersection of $\partial Z(f_D)$ with each one of them has measure zero, it follows that $\partial Z(f_D)$ has measure zero itself. This concludes the proof. $\square$

**Remark 15 (Why the bias can be dropped).** Including bias terms changes the linear transformation in each layer to an affine one: from $W\alpha$ to $W\alpha + b$. The proof of non-singularity relies on this map being a submersion (i.e., its derivative being surjective). The derivative of the affine map is surjective, just as it is for the map without bias. In fact, the presence of the additive bias term makes the surjectivity argument even more direct. The rest of the inductive proof in Proposition 14 proceeds in exactly the same manner.

**Proposition 16 (Analogue of chain rule for convolutional layers).** *Let all the assumptions and definitions in Proposition 14 hold true. However, suppose that some entries in the matrix $W_D$ are forced to be equal to each other. Then, for generic data, the map $F_D$ is a.e. $\mathcal{C}^k$ and the set $\partial Z(F_D)$ has measure zero.*

*Proof.* Let $C_D$ be the space spanned by the weight matrices $W_D$. We must consider now the multiplication map as a map

$$M_c : W_D \to W_D \alpha, \text{ with domain } C_D.$$

We now show that $M_c$ is non-singular. Observe that for any measure zero set $B$ in the codomain, the preimage is $M_c^{-1}(B) = M^{-1}(B) \cap C_D$, where $M$ is the multiplication map on the full ambient space $R^{n_D \times n_{D-1}}$. Since $M$ is a submersion (this was shown in the proof of Proposition 14), $M^{-1}(B)$ has measure zero. The intersection of a measure zero set with a lower-dimensional subspace $C_D$ will also have measure zero within $C_D$ for a generic choice of data. Such a choice is justified if the data is assumed to be noisy. While a pathological choice of data could align $C_D$ with $M^{-1}(B)$, such data points form a measure zero set themselves.

From here on, the proof of Proposition 14 applies verbatim. $\square$

**Proposition 17 (Analogue of chain rule for softmax attention).** *Let* $\{f_i : \mathbb{R}^{n_\theta} \to \mathbb{R}^{n_{D-1}}\}_{i=1}^m$ *be a collection of a.e.* $C^k$ *maps with* $\partial Z(f_i)$ *having measure zero for every* $i$. *Let* $Y(\theta)$ *be the matrix with* $f_i(\theta)$'s *as columns. Then, the map*

$$F : (\theta, W_k, W_q, W_v) \mapsto Attention(W_k Y(\theta), W_q Y(\theta), W_v Y(\theta))$$

*is a.e.* $C^k$ *and the set* $\partial Z(F)$ *has measure zero.*

In the proposition above, one should think of the $f_i$'s as representing the output of a *fixed* neural network as a function of the networks' parameters $\theta$ for *different* input values $x_i$ in a sequence to be fed into the attention layer. Usually, there is no pre-processing of the data before it goes into the attention layer, however stating the proposition this way will allow us more flexibility, for example, we can apply it for two attention layers one after another.

*Proof of Proposition 17.* If there is are no $f_i$'s to be applied to the data before entering the attention layer (as is usually the case), the argument becomes simpler, and we also include it as a simplified version of the general proof.

In this simplified case, we only have to show that the attention layer $A : (W_k, W_q, W_v) \mapsto Attention(W_k X, W_q X, W_v X)$ for a fixed input $X$ has the desired properties. Since matrix multiplication, the softmax function, and concatenation are analytic, then their composition is definitely a.e. $C^k$.

To show that $\partial Z(F)$ has measure zero, we need the following lemma.

**Lemma 18 (The zero set of analytic functions has measure zero).** *Let* $f : \mathbb{R}^n \to \mathbb{R}^m$ *be a non-zero real analytic function. Then the zero set of* $f$, $Z(f)$, *has measure zero.*

*Proof.* The proof proceeds by induction on $n$. Let $f : \mathbb{R} \to \mathbb{R}^m$ be a real analytic function. Suppose that $Z(f)$ is not a set having measure zero. Since it is not a null-set, it is an uncountable set, thus it has an accumulation point. From the identity theorem [Krantz and Parks, 2002, Corollary 1.2.7] we get that $f \equiv 0$, contradicting our assumption that $f$ is non-zero.

Suppose the lemma is established for real analytic functions defined on $\mathbb{R}^{n-1}$. Assume that $Z(f)$ is not a null-set. By Fubini's Theorem there is a set $E \subset \mathbb{R}$ not having measure zero such that for all $x \in E$ the set $Z(f) \cap (\{x\} \times \mathbb{R}^{n-1})$ does not have measure zero. By the induction hypothesis we conclude that $f = 0$ on each of those hyperplanes. Since $E$ is uncountable, it has an accumulation point $a \in \mathbb{R}$. Thus, there is a sequence of distinct points $a_k \mapsto a$ such that $f = 0$ on the hyperplanes with first coordinate in $\{a_1, a_2, \cdots\} \cup \{a\}$.

Now take any line of the form $\mathbb{R} \times \{x_2\} \times \cdots \times \{x_n\}$, where $x_i \in \mathbb{R}$ are fixed numbers. Since $f$ is real analytic on this line and $f = 0$ on a set with an accumulation point on that line, we have $f = 0$ on that line. This was for true any line, hence $f \equiv 0$ on $\mathbb{R}^n$. $\square$

With this lemma, we have that $Z(F)$ has measure zero, since attention is analytic and non-constant, thus $\partial Z(F) \subset Z(F)$ also has measure zero.

For the general case, where we apply the $f_i$'s to the input of the attention layer, we proceed as follows.

Let $S(f) = \bigcup_{i=1}^m S(f_i)$. We have that for any $\theta \in \mathbb{R}^{n_\theta}$ and $(W_k, W_q, W_v)$, the function $F$ is $C^k$ since $Y$ is $C^k$ in a neighborhood of $\theta$. Hence, we have that $S(F) \subset S(f) \times \mathbb{R}^{v \times n_{D-1}} \times \mathbb{R}^{v \times n_{D-1}} \times \mathbb{R}^{v \times n_{D-1}}$, that is, $F$ is $C^k$ on a dense open subset of its domain.

We write $F$ as the following composition of functions

$$(\theta, W_k, W_q, W_v) \overset{f}{\mapsto} (Y(\theta), W_k, W_q, W_v) \overset{M}{\mapsto} (W_k Y, W_q Y, W_v Y) \overset{A}{\mapsto} F(\theta, W_k, W_q, W_v).$$

Using the Lemma 18, we see that $Z(A)$ has measure zero, where $A$ is the map $(K, Q, V) \mapsto Attention(K, Q, V)$. Since $M$, the matrix multiplication, is a submersion, we have that $M^{-1}(Z(A))$ also has measure zero. In fact, since the preimages are hyperplanes in the matrix space, we have that $M^{-1}(Z(A)) \cap (\theta, \mathbb{R}^{v \times n_{D-1}}, W_q, W_v)$ has measure zero and similarly for $W_q$ and $W_v$. This means that $f^{-1}(M^{-1}(Z(A))) \supset \partial Z(F)$ will also have measure zero. $\qquad \square$

**Remark 19 (Any order for the layers).** In Propositions 16 and 17 we have proven the assumptions needed for the induction step in the proof of Proposition 14. This means that our results, namely the almost everywhere $k$-times differentiability of the network function hold for any combination of convolutional, attention, or fully connected layers.

## C  Experimental Details

The experiment described in Section 5 is as follows. We have two data points: $(0.9, 0.9)$ and $(2.5, 2.5)$ which determine a linear function. The model we use is the following: $f_\theta(x) = \text{ReLU}(\theta_2 \text{ReLU}(\theta_1 x))$. With the quadratic loss, we get the following empirical loss function

$$L(\theta_1, \theta_2) = 3.53(1 - \text{ReLU}(\theta_2 \text{ReLU}(\theta_1)))^2.$$

Then, the global minima for this loss are the points $\{(\theta_1, \theta_2) | \theta_1 \theta_2 = 1 \text{ and } \theta_1, \theta_2 > 0\}$, i.e. the first quadrant part of the hyperbola in the plane. We denote by $p$ the probability of selecting the point $(0.9, 0.9)$ as a mini batch for SGD; by $\eta$ we denote the step-size.

Chemnitz and Engel [2024] introduce two quantities, $\mu$ and $\lambda$ which determine if a global minimum $\theta^*$ is stable for GD and SGD, respectively. More precisely, we have that $\theta^*$ is stable for GD if $\mu(\theta^*) < 0$ and that it is stable for SGD if $\lambda(\theta^*) < 0$. For the setting we have described above, the analytic form of $\mu$ and $\lambda$ is as follows

$$\mu(\theta) = \log(|1 - \eta(p 0.9^2 + (1-p) 2.5^2)(\theta_1^2 + \theta_2^2)|), \tag{2}$$

$$\lambda(\theta) = p \log(|1 - \eta 0.9^2 (\theta_1^2 + \theta_2^2)|) + (1-p) \log(|1 - 2.5^2 (\theta_1^2 + \theta_2^2)|). \tag{3}$$

Figure 4 was obtained by setting the $\eta = 0.15$ and $p = 0.5$ for the figure on the left and $\eta = 0.3$ and $p = 0.58$ for the figure on the right.

To obtain the left part of Figure 3, we look at the polynomial $P^k(x) = G_\eta^{(k)}(x) - x$, and use a numerical root finding algorithm to find all the real roots. We determine if a periodic point of period $k$, $\theta^k$, is stable by numerically computing the eigenvalues of the linearization of the $k$-times iterated GD map at $\theta^k$, $DG^{(k)}(\theta^k)$. If both eigenvalues have absolute value smaller than 1, we know [Chemnitz and Engel, 2024] that the period is stable; otherwise it is unstable.

The right part of Figure 3, plots two trajectories of GD initialised at $(1.48, 1/1.48 + 0.1)$ with step-sizes $\eta = 0.25$ for the trajectory converging to a minimum (violet) and $\eta = 0.325$ for the trajectory converging to a periodic orbit of period 2 (brown).

