# OpenReview forum: "Non-Singularity of the Gradient Descent Map for Neural Networks with Piecewise Analytic Activations"
_NeurIPS.cc/2025/Conference — NeurIPS 2025 poster_

### Official Review · Reviewer_9vB6 · 2025-06-24

**Clarity:** 3
**Significance:** 3
**Originality:** 3
**Rating:** 4
**Confidence:** 4

**Summary:**

The paper proves that, for neural networks with piece-wise analytic activations (ReLU, sigmoid, etc.), the gradient-descent map (GD map) is non-singular (it does not turn a measure-zero set into a set with positive measure) for neural networks with piece-wise analytic activations and a fixed learning rate $\eta$. The proof uses standard real-analytic arguments and covers fully connected and convolutional nets. A 2-D toy example is shown.

**Questions:**

- Can you give a concrete bound or heuristic for choosing $\eta$ outside the measure-0 “bad’’ set?

**Ethical Concerns:**

["NO or VERY MINOR ethics concerns only"]

**Final Justification:**

I agree that the paper’s rigorous proofs and the new layered chain-rule lemma provide a somewhat meaningful theoretical advance. Although the experimental validation is limited and the paper does not offer precise guidance on choosing η, these shortcomings are not fatal. Thus, a “4 – Borderline Accept” recommendation is reasonable.

**Limitations:**

Yes

**Paper Formatting Concerns:**

Appendix is should be separated as a supplementary document.

**Quality:**

3

**Strengths And Weaknesses:**

Strengths
- Clear mathematics and complete proofs.
- This paper revises an existing theoretical gap for ReLU-type networks.

Weaknesses
- Result has little practical impact because, on bounded parameter sets, losses are already Lipschitz and earlier non-singularity arguments apply.
- Although the experimental results is helpful for showing the theory presented in this paper, in practical setting the learning rate changes (decays) during the learning.  The experiment provides little implication to practical situation in machine learning.

---

> ### Author Rebuttal · Authors · 2025-07-30
>
> We sincerely thank the reviewer for their time and for their assessment that our paper features “clear mathematics and complete proofs" and "revises an existing theoretical gap." We appreciate the opportunity to clarify the practical impact of our work and answer the questions raised.
>
> ### **On the Practical Impact and Boundedness of Parameters**
> *Critique: "Result has little practical impact because, on bounded parameter sets, losses are already Lipschitz and earlier non-singularity arguments apply."*
>
> Thanks for raising this important point, which allows us to highlight a key motivation for our work. While it is true that prior non-singularity results apply to Lipschitz-smooth losses, the assumption that GD trajectories remain in a bounded set where the loss is Lipschitz is itself a strong condition that lacks theoretical grounding. To our knowledge, no existing results guarantee that the GD trajectory will remain bounded without a priori assuming Lipschitz smoothness and a sufficiently small step-size. Relying on the empirical observation that successful training runs stay bounded can be a form of survivorship bias, where initializations and hyperparameters leading to divergence are simply discarded.
>
> Our work's practical significance lies in removing this restrictive assumption of Lipschitzness (or, equivalently, of bounded parameter sets). We provide the first non-singularity guarantee that holds even when the trajectory is not confined to a bounded region. This is particularly relevant to modern training paradigms like "Edge of Stability" (EoS) training, where networks are intentionally trained with large learning rates that violate the conditions for the Gradient Descent Lemma (used to prove convergence). Our result provides the foundational theoretical justification on top of which other works built in order to explain why optimization in these practical, high-performance regimes does not fail, thus bridging a critical gap between theory and practice.
>
> ### **On the Implications for Varying Learning Rates**
> *Critique: "Although the experimental results are helpful for showing the theory presented in this paper, in practical setting the learning rate changes (decays) during the learning."*
>
> This is an excellent point, and we want to clarify that our framework directly extends to settings with varying learning rates, such as those with schedules or adaptive methods.
>
> Our main result (Theorem 1) establishes that the GD map, $G_\eta$, is non-singular for almost all step-sizes $\eta$. An optimization process over $T$ steps with a learning rate schedule $(\eta_1, \eta_2, \dots, \eta_T)$ can be described by the composite map $G = G_{\eta_T} \circ \dots \circ G_{\eta_1}$. A fundamental property of non-singular maps is that their composition is also non-singular. Since each individual map $G_{\eta_t}$ in the sequence is non-singular (with probability one), their composition $G$ is also guaranteed to be non-singular.
>
> Therefore, our analysis provides a rigorous foundation for understanding optimization dynamics not just for a fixed step-size, but for any practical scheme that relies on composing gradient descent steps. The same reasoning applies if we also use different data for the batches at each step of the optimization, illustrating how Theorem 1 also extends to SGD as well. We will add a sentence to the Discussion to make this important implication more explicit.
>
> ### **On Choosing a "Good" Learning Rate**
> *Critique: "Can you give a concrete bound or heuristic for choosing η outside the measure-0 “bad’’ set?"*
>
> Before we answer, we have to clarify what it means to have a “good” or a “bad” learning rate. In this work, a “bad” learning rate is one out of a measure zero set for which the GD map is singular; **a “good” learning rate is one that is not “bad”, i.e., one for which the GD map is non-singular**. With this clarification, our theoretical result provides a strong and practical heuristic. The proof of Proposition 11 shows that a step-size $\eta$ is "bad" (i.e., GD map is singular) only if 1/$\eta$ is precisely an eigenvalue of the loss Hessian. Our work establishes that the set of all such "bad" values has Lebesgue measure zero. In practice, this means that the probability of selecting a "bad" step-size by chance is zero for any selection method that is absolutely continuous with respect to the Lebesgue measure (this covers common heuristics). Schedules or adaptive methods are also included, since the probability of sampling a finite number of learning rates from the “bad” ones is still zero. Any other criterion for labeling a step size as “good” does not fall within the scope of this work.
>
> ### **On the Novelty of the Proof Technique**
> Finally, we wish to gently counter the characterization of our proof as using "standard real-analytic arguments." As we detail in Remark 5, standard tools like the chain rule fail for the composition of almost everywhere analytic functions, which is the central challenge when moving beyond simple analytic activations. Our key technical innovation, presented in Proposition 6, is a novel "analogue of the chain rule" that leverages the specific layered structure of neural networks to overcome this limitation. To our knowledge, this inductive proof technique is a new contribution to the literature

---

> ### Comment · Reviewer_9vB6 · 2025-08-05
> **Thank you for your response.**
>
> Thank you for your response.  I've updated my score.

---

### Official Review · Reviewer_bN8y · 2025-06-27

**Clarity:** 2
**Significance:** 2
**Originality:** 2
**Rating:** 4
**Confidence:** 4

**Summary:**

This paper advances the theoretical understanding of gradient descent (GD) optimization in deep neural networks by proving the non-singularity of the GD map, a key assumption for convergence to global minima and avoidance of saddle points. Unlike prior studies that assume non-singularity or rely on restrictive conditions like Lipschitz smoothness or small step-sizes, the authors demonstrate that the GD map is non-singular for almost all step-sizes in realistic neural network architectures, including fully connected, convolutional, and softmax attention layers with piecewise analytic activations (e.g., sigmoid, ReLU, leaky ReLU). The proof first demonstrates that neural network mappings are almost everywhere analytic by leveraging an analogue of the chain rule for neural networks. This property is then used to analyze the measure of preimages under the GD map, establishing its non-singularity.

**Questions:**

1. Extending Non-Singularity to SGD: The paper claims the non-singularity of the gradient descent (GD) map extends to stochastic GD (SGD), but this is not convincingly supported. Extending GD results to SGD typically requires modeling randomness, introducing significant theoretical complexity. Could the authors provide a rigorous analysis or additional assumptions to justify this extension?
2. Clarifying the Almost Everywhere Analytic Proof: The proof that neural network mappings are almost everywhere analytic lacks clarity and rigor, particularly for architectures like attention modules, where the appendix provides only brief descriptions without detailed derivations. Additionally, the analysis seems tailored to supervised learning; its applicability to generative model training paradigms is unclear. Could the authors elaborate on the proof for diverse architectures and discuss its relevance to generative models?
3. Connecting Applications to Theoretical Results: The Discussion section’s applications—periodic trajectories and stable minima of GD versus SGD—feel disconnected from the main non-singularity proof and lack novel insights compared to prior work. Could the authors clarify how these applications leverage the proved results to offer new perspectives or contributions?
4. Significance of Unrestricted Step-Sizes: The proof emphasizes non-singularity for arbitrary step-sizes, but the practical significance of this claim is unclear. Traditional optimization theory suggests that step-sizes exceeding twice the inverse of the Hessian’s maximum eigenvalue lead to divergence.

**Ethical Concerns:**

["NO or VERY MINOR ethics concerns only"]

**Final Justification:**

Although I believe there are still some points that the paper can optimize as I listed in my comments to authors' rebuttal, I think the paper is ready to be published at NeurIPS. I thus increase my rating to 4.

**Limitations:**

1. The use of strong analyticity to prove non-singularity is not justified when weaker smoothness conditions suffice for convergence.
2. The applicability to stochastic GD lacks analysis of randomness-related complexities.
3. The proof’s relevance to generative model training is unclear and unaddressed.
4. The Discussion’s applications (periodic trajectories, stable minima) lack novelty and connection to the main results.

**Paper Formatting Concerns:**

None.

**Quality:**

3

**Strengths And Weaknesses:**

__Strengths__
1. The paper rigorously proves that the gradient descent (GD) map is non-singular for almost all step-sizes in realistic neural network architectures. The proof is clearly presented without apparent theoretical flaws. The core argument begins with establishing the almost everywhere analytic property of neural network mappings. The intuition is that common model components are almost everywhere analytic, with non-analytic points forming measure-zero sets in each layer. Through the composition of these layers, the non-analytic set in the entire parameter space remains measure-zero. Building on this, the authors analyze the measure of preimages in critical and non-critical regions to derive the non-singularity of the GD map. The work appears complete, with a well-structured proof addressing the stated problem.
2. The significance of the result is tempered by the fact that the strong condition of analyticity—requiring functions to be infinitely differentiable with convergent power series expansions—is used to prove a relatively weaker conclusion of non-singularity. Prior work has directly leveraged analyticity to establish stronger results, such as GD convergence to global minima, while other convergence analyses (e.g., those using the Polyak-Łojasiewicz condition) rely on weaker smoothness assumptions.
3. Prior work has extensively analyzed the analytic properties of neural networks, particularly for piecewise analytic activations. For instance, studies have shown that neural networks with ReLU activations are almost everywhere analytic. To the best of our knowledge, using the almost everywhere analytic property to prove the non-singularity of the gradient descent (GD) map is novel, offering a new perspective on the theoretical underpinnings of GD optimization in neural networks.

__Weakness__:
1. The paper is generally well-organized, with a clear logical progression in presenting the proof of the gradient descent (GD) map’s non-singularity, structured around the stepwise demonstration of the core theorem. However, the writing is somewhat difficult to follow due to insufficient explanations of key terms, variables, and the intuitions or practical implications behind the proved lemmas and theorems.
2. In the section proving that neural network mappings are almost everywhere analytic, the analysis for certain architectures, such as attention modules, lacks rigor in the appendix. The appendix briefly describes the operations and functions of these modules but does not provide a detailed derivation or proof of their analytic properties
3. The Discussion section presents two applications—periodic trajectories and stable minima of GD versus SGD—that feel disconnected from the main theoretical analysis. These applications address simplistic cases, lack clear connections to the paper’s core results, and do not demonstrate novel contributions compared to existing work on these topics.

---

> ### Author Rebuttal · Authors · 2025-07-30
>
> We thank the reviewer for recognizing that our work "advances the theoretical understanding of gradient descent" and that our core proof is "novel, offering a new perspective."
>
> ### **On the Applicability to Stochastic Gradient Descent (SGD)**
> *Critique: "The paper claims the non-singularity of the gradient descent (GD) map extends to stochastic GD (SGD), but this is not convincingly supported."*
>
> We apologize if our justification was not sufficiently clear. Our theoretical framework robustly covers the SGD setting. The core of our proof, culminating in Theorem 1, establishes non-singularity for *any fixed data and any batch size*. An SGD step is mathematically equivalent to a full GD step performed on a loss function computed over a mini-batch of data. Since our result holds *irrespective* of the number of data points used to compute the loss (as stated on lines 68-70 and 298-304), it directly applies to each step of an SGD trajectory, where the data changes at each iteration. The map for each step, $G_{\eta,i}$, is therefore non-singular (for almost all $\eta$), and the composition of these maps, which describes the full SGD trajectory, is also non-singular. We will revise the text to make this crucial point more explicit and prominent.
>
> ### **On the Rigor of the Proof for Different Architectures**
> *Critique: "The proof that neural network mappings are almost everywhere analytic lacks clarity and rigor, particularly for architectures like attention modules... its applicability to generative model training paradigms is unclear."*
>
> We thank the reviewer for pushing for greater clarity.
> * **Attention Layers**: The proof for attention layers (Proposition 15) relies on a straightforward but powerful observation. The attention mechanism itself, as a map from parameters $(W_q, W_k, W_v)$ and a fixed input matrix $X$, is a fully analytic function. Even though the attention mechanism is rather complicated, *it is made up only of analytic functions*. The almost everywhere analyticity of the full network map arises from the composition of this analytic attention map with the a.e. analytic output of the following layers. As established in Lemma 8, the composition of an analytic function with an a.e. analytic function is itself a.e. analytic. We will expand the appendix to make this argument more self-contained.
> * **Generative Models**: This is an excellent suggestion. While we did not explicitly analyse generative models, our framework is potentially applicable. Many generative training paradigms involve optimizing a loss function. If this loss can be shown to be a.e. analytic with respect to the generator's parameters, our non-singularity results apply. Since we already show that the functions determined by the models are a.e. analytic, all that is left is to show that the specific losses used for generative modelling will preserve this. If the losses are analytic, an argument similar to what we have above, using Lemma 8 for composition of functions, will hold and the GD map will be non-singular. We will add a brief comment on this as a promising direction for future work in the Discussion section.
>
> ### **On Connecting the Discussion Section to the Main Result**
> *Critique: "The Discussion section’s applications—periodic trajectories and stable minima of GD versus SGD—feel disconnected from the main non-singularity proof..."*
>
> We appreciate this feedback and will revise the Discussion to make the connection more explicit. The applications presented are direct consequences of our main theorem. The stability analysis of periodic trajectories (Figure 3) and the comparison of stable minima between GD and SGD (Figure 4) both rely on analyzing the behavior of the iterated GD map ($G_\eta^k$) and the SGD map ($G_{\eta,i}$), respectively. The non-singularity of these maps, which is guaranteed by our theorem and the property that compositions of non-singular maps are non-singular, is the foundational requirement that makes such stability analyses theoretically sound. Our work provides the missing justification for applying these analytical tools to practical neural networks.
>
> ### **On the Significance of Unrestricted Step-Sizes**
> *Critique: "The proof emphasises non-singularity for arbitrary step-sizes, but the practical significance of this claim is unclear."*
>
> We first emphasise that **the scope of this work was to show that the GD map is non-singular, not to look at other aspects of the optimisation process such as convergence**. This is a crucial point that underscores the significance of our work. Traditional optimization theory (e.g., Lee et al. (2019)) proves saddle-point avoidance under the strong assumption of Lipschitz smoothness, which in turn requires small step-sizes. However, this assumption is violated in many practical, high-performance training regimes. Mohtashami, Jaggi and Stich (2023), for instance, shows that networks are often best trained with large learning rates where the loss is demonstrably not Lipschitz smooth.
>
> Our result is significant precisely because it does not require small step-sizes or Lipschitz smoothness. By proving non-singularity for almost all step-sizes, we provide the first rigorous theoretical foundation that helps explain the success of these empirically validated, aggressive training strategies. Thus, our work allows for the extension of relevant guarantees from the optimisation literature to the practical neural networks setting as well. We will revise the introduction and discussion to more strongly emphasise this important connection.
>
> ### **On the Generality of Our Work**
> We agree that prior works have used analyticity to prove stronger results, such as convergence. However, the analyses *do not hold a priori* for the piecewise analytic setting. This was a key motivation for our work, which aims to bridge this gap. Our contribution is to establish the non-singularity of the GD map in this more general piecewise analytic setting (e.g. for networks with ReLU activation). This result is a foundational prerequisite that enables the rich body of convergence analyses to be extended to these more practical network architectures.
>
> ### **On the Scope of Our Claims**
> To clarify, our paper does not claim to prove convergence. Our focus is specifically on proving the non-singularity of the gradient descent map. We view this as a critical intermediate step that validates a key assumption used in many convergence analyses. By establishing this property, our work provides the necessary tools for future research to carry over convergence results to the piece-wise analytic setting.
>
> ### **On the Polyak-Łojasiewicz (PL) Condition**
> Regarding the comparison to the PL condition, we believe there may be a misunderstanding of the relationship between these concepts. Piecewise analyticity is more akin to a structural property, whereas the PL condition is a geometric property. These are distinct and non-hierarchical assumptions; a function can be piecewise analytic without satisfying the PL condition, and vice-versa. Our work focuses on the structural properties stemming from the network’s architecture, which is a different and complementary line of analysis. We will add a brief note to the related works section to clarify this distinction.
>
> References:
> * Mohtashami, A., Jaggi, M., and Stich, S. 2023. Special Properties of Gradient Descent with Large Learning Rates.
> * Lee, J. D.; Panageas, I.; Piliouras, G.; Simchowitz, M.; Jordan, M. I.; and Recht, B. 2019. First-order methods almost always avoid strict saddle points.

---

> > ### Comment · Reviewer_bN8y · 2025-08-05
> >
> > Thanks to the authors for clarifying my earlier misunderstandings—especially the extension to SGD as a composition of per-mini-batch GD maps and the scope distinction between structural (piecewise analytic) and geometric assumptions. I would be willing to raise my score to 4.
> >
> > Besides, I believe this paper still has the following points that can be optimized:
> > 1. Rigor and Scope for Different Architectures (attention/masks/MoE).
> > The paper asserts almost-everywhere analyticity for common modules. However, the architectural scope still reads as broader than what the arguments transparently cover.
> > - **Masks.** Input-dependent sparsification (e.g., top-k/thresholded attention) introduces discrete changes and non-smooth boundaries; these are not analytic (often not even continuous).
> > - **MoE hard routing.** Argmax/top-k expert selection is a discrete operation; the induced mapping is non-analytic and piecewise constant on regions separated by routing boundaries.
> >
> > 2. Discussion Section’s Connection to the Main Result.
> > The Discussion fails to establish a formal bridge from the non-singularity theorem to the two applications (periodic trajectories; GD vs. SGD stable minima): it neither instantiates the theorem’s hypotheses (piecewise analyticity, “almost all” step sizes, non-singular GD map) in the toy setups of Figs. 3–4 nor shows how the theorem’s conclusion is used in those analyses; thus, while it claims foundational support for the underlying assumptions, the presented examples show no operative link to the theorem.

---

> ### Author Response · Authors · 2025-08-05
>
> We are happy that our initial clarifications were helpful.
>
> ### **On the Rigor and Scope for Different Architectures**
> *Critique: "…the architectural scope still reads as broader than what the arguments transparently cover..."*
>
> Our analysis covers architectures where the constituent operations are piecewise analytic. This includes the standard and most widely used attention mechanism introduced by Vaswani et al. (2017). We agree that architectures employing discrete, non-differentiable routing or selection mechanisms—such as hard attention with argmax, top-k sparsification, or Mixture-of-Experts (MoE) models with hard routing—are fundamentally different (Bergsträßer et al. (2024)). Our analysis is focused specifically on the non-singularity of the *gradient descent map*. Architectures employing discrete operations are often trained with different, specialized optimization schemes (e.g., reinforcement learning-based methods (Nikpour and Armanfard (2023)) or those using the Gumbel-Softmax trick (Jang et al. (2017)) precisely because standard gradient descent is not directly applicable. As such, these architectures fall outside the defined scope of our study. If the reviewer could provide any specific references, where gradient descent based optimisation is used for hard/threshold attentions, MoE, etc., we could then provide a more specific response.
>
> To prevent any ambiguity for future readers, we will revise the paper to explicitly state this boundary. We will add a sentence to the Discussion section clarifying that our results apply to architectures amenable to standard gradient-based optimization and that models with discrete, non-differentiable components would require a separate non-singularity analysis tailored to their specific optimization algorithms.
>
> ### **On the Connection Between the Discussion Section and the Main Result**
> *Critique: "The Discussion fails to establish a formal bridge from the non-singularity theorem to the two applications..."*
>
> We will add the following explanation to the Discussion section (before the paragraph starting at line 313):
>
> “To illustrate the connection between our result and the stability of minima, we first recall a key result from Chemnitz and Engel (2024). They show that for a generic parametric model, if the GD map is non-singular and the set of global minima $M$ forms a submanifold, then the stability of a minimum $\theta \in M$ is determined by a computable quantity. Our Theorem 1 provides the missing piece to rigorously apply this framework to neural networks. By establishing the non-singularity of the (S)GD map for neural networks with piecewise analytic activations, we can now state the following direct corollary:
>
> **Corollary.** *Let the assumptions be as in Theorem 1. Additionally, suppose the data is generic and let $M$ be the set of global minima of the loss function $L$. Then, for almost all step-sizes, there exist functions $\mu(\theta)$ and $\lambda(\theta)$ such that a global minimum $\theta \in M$ is stable under GD or SGD only if $\mu(\theta) < 0$ or $\lambda(\theta) < 0$, respectively. Conversely, if $\mu(\theta) > 0$ or $\lambda(\theta) > 0$ for a global minimum $\theta$, then $\theta$ is unstable.*
>
> The definitions of $\mu$ and $\lambda$ are the same as those in Chemnitz and Engel (2024). For a discussion of “generic data”, we refer the reader to Cooper (2021); note that it is reasonable to assume that noisy data is generic.
>
> We now use this corollary to analyze the training dynamics in our examples. We first compute the functions $\mu$ and $\lambda$ for the simple setting with loss function $(\theta_1, \theta_2) \mapsto 3.51(1-\text{ReLU}(\theta_2\text{ReLU}(\theta_1)))^2$ (corresponding to a two-layer ReLU neural networks as described in Appendix C). We then use them to: 1) determine the stability of periodic trajectories under GD as the step-size $\eta$ changes (Figure 3), and 2) show that for a fixed step-size, a global minimum can be stable under GD but unstable under SGD, and vice-versa (Figure 4).”
>
> We believe this addition, along with a corresponding update to Appendix C explicitly verifying that the experimental setup satisfies the hypotheses of Theorem 1, will fully address the reviewer's concern.
>
> References:
> Ashish Vaswani, Noam Shazeer, Niki Parmar, Jakob Uszkoreit, Llion Jones, Aidan N. Gomez, Łukasz Kaiser, and Illia Polosukhin. Attention is all you need. NIPS’17.
>
> Pascal Bergsträßer, Chris Köcher, Anthony Widjaja Lin, and Georg Zetzsche. The Power of Hard Attention Transformers on Data Sequences: A formal language theoretic perspective. NIPS’24.
>
> Bahareh Nikpour and Narges Armanfar. Language-Guided Reinforcement Learning for Hard Attention if Few-Shot Learning. arXiv:2310.07800v3.
>
> E. Jang, S. Gu, and B. Poole. “Categorical Reparameterization with Gumbel-Softmax.” ICLR, 2017.
>
> Dennis Chemnitz and Maximilian Engel. Characterizing dynamical stability of stochastic gradient descent in overparameterized learning. arXiv:2407.20209.

---

> > ### Comment · Reviewer_bN8y · 2025-08-06
> >
> > Thank you for your clarification. I think the paper would be more solid if these adjustments could be reflected in the latest version.

---

### Official Review · Reviewer_41PX · 2025-07-01

**Clarity:** 3
**Significance:** 3
**Originality:** 3
**Rating:** 5
**Confidence:** 4

**Summary:**

The authors prove that for almost all learning rates the function that applies a gradient descent step to the parameters of a neural network never maps sets of positive measure to null sets, which has been assumed in many theoretical works but without a formal proof so far.

**Questions:**

Can you justify that the assumption needed for Proposition 14 to work is realistic? In any case, this assumption should ideally be specified in Proposition 7.

What exactly does "different probabilities for sampling mini-batch data" mean in the caption of Figure 4? Does the expectation of the minibatch gradient equal the full batch gradient?

The definition of analiticity in lines 185-186 is not correct. The set $N(f_\theta)$ in Figure 2 is defined only in the appendix. The letter $H$ is used both for the depth of the network and for the Hessian of the loss, which is a bit confusing. It would help the reader if the references were ordered alphabetically.

Some typos: line 197 complemets -> complements, line 340 networks -> network, line 350 of recurrent -> of a recurrent.

**Ethical Concerns:**

["NO or VERY MINOR ethics concerns only"]

**Final Justification:**

The authors properly addressed the issues I raised in the rebuttal. My main concern was the problem with the result concerning convolutional layers, which seems to be solved. Therefore I increase the rating from 4 to 5.

**Limitations:**

yes

**Quality:**

3

**Strengths And Weaknesses:**

1. Quality: I believe the main results and the corresponding proofs are correct, except the treatment of convolutional neural networks. Proposition 14 in Appendix B assumes that the matrices corresponding to the linear maps that are induced by convolution kernels span the whole space, which seems like an unrealistic assumption and is not mentioned in Proposition 7. The experiment with its results depicted in Figure 4 on the right used "different probabilities for sampling mini-batch data", which sounds like a violation of the requirement that the expectation of the minibatch gradient should equal the full batch gradient.

2. Clarity: The paper is well-written, paints a nice picture of relevant works in which the results aim to fill an important whole and the proof ideas are communicated clearly.

3. Significance: The results are already assumed by many works and therefore are expected to be built upon by the community.

4. Originality: While the results were expected to hold, they do fill an important gap in the theory of deep learning. Moreover, the proof techniques can potentially lead to obtaining additional results later on.

---

> ### Author Rebuttal · Authors · 2025-07-30
>
> We are grateful to the reviewer for their careful and constructive feedback. We are especially thankful for the identification of a subtle error in our proof, which has allowed us to strengthen the paper's technical correctness.
>
> ### **On the Assumption for Convolutional Layers (Proposition 14)**
> *Critique: "Can you justify that the assumption needed for Proposition 14 to work is realistic? In any case, this assumption should ideally be specified in Proposition 7."*
>
> We sincerely thank the reviewer for identifying this issue. Upon reflection, we agree that the assumption in the original Proposition 14—that the matrices corresponding to convolution kernels span the entire ambient space—is not realistic. We have corrected our argument, which we present here and will incorporate into the final version. The claim that convolutions can be represented by matrices with some entries equal to each other can also be found in Chapter 9.1 from Goodfellow et al. (2016).
>
> The core of the proof remains the same, but we no longer need the unrealistic assumption that the matrices span the ambient space. For a convolutional layer, the weight matrices $W_h$ with equal entries do not span the full matrix space $\mathbb{R}^{d_h \times d_{h-1}}$, but rather a smaller linear subspace, let's call it $C_h$. The multiplication map must now be considered as a map $M_c: W_h \to W_h \alpha$ with domain $C_h$.
>
> To show $M_c$ is non-singular, we observe that for any measure-zero set $B$ in the codomain, the preimage is $M_c^{-1}(B) = M^{-1}(B) \cap C_h$, where $M$ is the multiplication map on the full ambient space. Since $M$ is a submersion for almost all data $\alpha$, its preimage $M^{-1}(B)$ has measure zero. The intersection of a measure-zero set with a lower-dimensional subspace $C_h$ will also have measure zero within $C_h$ for a generic choice of data. Such a choice is justified if the data is assumed to be noisy. While a pathological choice of data could align $C_h$ within $M^{-1}(B)$, such data points form a measure-zero set themselves (an argument inspired by the genericity results in Cooper (2021)).
>
> We will therefore revise Proposition 14 to read: "Let all assumptions and definitions in Proposition 13 hold true. However, suppose that some entries in the matrix $W_h$ are forced to be equal to each other. Then for generic data, the map $F_h$ is a.e. $C^k$ and the set $\partial Z(F_h)$ has measure zero."
>
> ### **On the Experimental Details in Figure 4**
> *Critique: "What exactly does 'different probabilities for sampling mini-batch data' mean in the caption of Figure 4? Does the expectation of the minibatch gradient equal the full batch gradient?"*
>
> We apologize for the confusing phrasing. The experiment considers two different data-generating distributions. In each case, the process produces only two points, $x_1$ and $x_2$, with probabilities $p$ and $(1-p)$, respectively. We train with a batch size of $1$. Therefore, $p$ simultaneously defines the underlying data distribution and the probability of sampling a given point. For any fixed $p$, the expectation of the mini-batch gradient is indeed equal to the full-batch gradient for that specific data distribution. The two plots in Figure 4 correspond to two different experiments with different underlying data distributions (i.e., different values of $p$). We will revise the caption in Figure 4 to make this distinction clear.
>
> ### **On Minor Corrections**
> *Critique on: definition of analyticity, set definitions in Figure 2, notation for depth ($H$), and typos.*
> * We will correct all of these points in the final version:
> * We will amend the definition of analyticity to explicitly state that the Taylor series must converge to the function.
> * We will add the formal definitions for the sets in Figure 2 to the main body of the text.
> * We will change the notation for network depth to avoid collision with the Hessian.
> All typos will be corrected, and we will re-order the references alphabetically.
>
> References:
> * Ian Goodfellow, Yoshua Bengio, and Aaron Courville. Deep Learning. MIT Press, 2016.
> * Yaim Cooper. Global minima of overparameterized neural networks. SIAM Journal on Mathematics of Data Science, 3(2):676–691, 2021.

---

> > ### Comment · Reviewer_41PX · 2025-08-02
> >
> > Thank you for addressing my concerns. I update my review accordingly.

---

### Official Review · Reviewer_NUng · 2025-07-01

**Clarity:** 4
**Significance:** 4
**Originality:** 3
**Rating:** 5
**Confidence:** 4

**Summary:**

Review: Non-Singularity of the Gradient Descent Map for
Neural Networks with Piecewise Analytic Activations

1) importance of problem: The paper addresses a fundamental, open theoretical problem—the non-singularity of the gradient descent map—which underpins many theoretical results in optimization, convergence, and stability analyses.

2) novelty of methods/approach.The approach was a clear one using proof techniques that offer clarity. ( e.g rather than submresion the paper could have used Morse theory or Sard’s theorem could also have been applied to achieve similar results. Sard’s theorem, in particular, provides another way to argue measure-zero singularities by showing critical values form a measure-zero set). I am not sure about the novelty of the methods, as these are still standard techniques.

3) significance of results: The assumption of non-singularity of gradient descent is a common one, so this proof is validating. However, many works have proven related results, and it was unlikely for the assumption to have been a false one.

4) clarity of presentation: The paper was cleary leaving obscuring technical details to the appendix

5) technical correctness: It’s as technically sound as I can tell given the amount of time I’ve had to read it

**Questions:**

see above.

**Ethical Concerns:**

["NO or VERY MINOR ethics concerns only"]

**Final Justification:**

The paper addresses a fundamental, open theoretical problem—the non-singularity of the gradient descent map—which underpins many theoretical results in optimization, convergence, and stability analyses.	 Accept the paper.

**Limitations:**

see above.

**Paper Formatting Concerns:**

Typos

Line 187: "almost eveywhere analytic functions"
Should be: "almost everywhere"
Line 197: "whose complemets have measure zero"
Should be: "complements"
Line 80: "to better to understand how the layered structure"
Should be: "to better understand"
Line 283: "Kato, 1995]. ¯"
The macron symbol (¯) appears to be a formatting error and should be removed
Line 428: "Tosio Kato.¯ Perturbation theory"
Again, the macron symbol (¯) should be removed

**Quality:**

3

**Strengths And Weaknesses:**

Strengths

The primary objective of this paper is to generalize existing theoretical results to neural networks using popular activation functions such as ReLU, sigmoid, and leaky ReLU. The authors clearly show  that the gradient descent (GD) map is not singular for almost all step sizes, which extends the range of existing theoretical frameworks. This generalization is backed up by:

1. Analyticity Results: The authors show that neural networks with piecewise analytic activations  are analytic almost everywhere (a.e.). The method  uses structural induction based on the layered structure of neural networks.

2. Non-Singularity Analysis: The authors use submersion theory and measure theory to rigorously show that almost all step-sizes are non-singular, using Kato's theorem to do so.

3. Versatility: The results apply to a wide range of network architectures, such as fully connected, convolutional, and attention layers.


Critique

1. Bias Simplification:  The authors set biases to zero as a simplifuing assumption. They claim simplification is easy to generalize, but they don't give a convinceing reason for that claim. This omission could make their findings less useful for a wide range of situations, since biases are used in all practical neural networks. More should be said about this claim in the body or the full proof should just be given in the appendix.

2. Attention Layers: The paper's discussion of attention mechanisms (Proposition 15) is short compared to how well it explains fully connected and convolutional layers. Because attention layers use complicated nonlinear operations, a more detailed and thorough explanation of why non-singularity should apply to attention mechanisms would be good.

3.  Practical Significance of "Almost Everywhere" Conditions: The use of measure-zero singular sets, while mathematically sound, lacks a detailed discussion regarding its practical implications. It's not clear if these theoretically unimportant singularities could have a big effect on  optimization problems, especially when there are a lot of parameters. For example what about non analytic points on the boundary of analytic regions?

5. Singular Step-Size Considerations: The authors only briefly talk about singular step-sizes (where η = 1/λᵢ). A more in-depth look at how these unique step sizes might affect adaptive step-size methods that are often used in neural network training would make the results more useful in real life.

---

> ### Author Rebuttal · Authors · 2025-07-30
>
> We are grateful for the reviewer's positive assessment, and for recognizing the significance of our work.
>
> ### **On the Bias Simplification**
> *Critique: "The authors set biases to zero as a simplifying assumption. They claim simplification is easy to generalize, but they don't give a convincing reason for that claim."*
>
> This is a fair point. Including bias terms changes the linear transformation in each layer to an affine one: from $W\alpha$ to $W\alpha + b$. The proof of non-singularity relies on this map being a submersion (i.e., its derivative being surjective). The derivative of the map $(W, b) \mapsto W\alpha + b$ is surjective, just as it is for the map without bias. In fact, the presence of the additive bias term makes the surjectivity argument even more direct. The rest of the inductive proof in Proposition 13 proceeds in exactly the same manner. We will add a paragraph to the appendix to formalize this argument and remove any ambiguity.
>
> ### **On the Discussion of Attention Layers**
> *Critique: "The paper's discussion of attention mechanisms (Proposition 15) is short compared to how well it explains fully connected and convolutional layers."*
>
> The proof for attention layers is indeed concise because the core argument is quite direct. The attention mechanism itself is an analytic function of its parameters $(W_q, W_k, W_v)$ for a fixed input — it consists of matrix multiplication, followed by an application of a softmax function. While attention is complicated, all of the operations involved are analytic, therefore they do not pose any challenges to our work. The almost everywhere analyticity of the full network map is therefore a direct consequence of Lemma 8: composing the analytic attention map with the a.e. analytic output of the preceding layers results in an a.e. analytic map. We will revise the text to better connect the proof in Proposition 15 with the discussion in Appendix A to make the argument more self-contained and clear.
>
> ### **On the Practical Significance of "Almost Everywhere" Conditions**
> *Critique: "The use of measure-zero singular sets...lacks a detailed discussion regarding its practical implications."*
>
> The practical significance is profound: if the sets of non-analytic or singular points were "thick" (i.e., had positive measure), then standard gradient-based optimizers would frequently fail or become trapped. In such a scenario, one would need to resort to more complex and computationally expensive methods, such as subgradient descent, which are also more difficult to analyze theoretically. Our result, by proving that these pathological sets are of measure zero, provides the rigorous justification for why the simpler, more efficient, and universally used first-order methods like GD and SGD are so effective in practice.
>
> ### **On Singular Step-Size Considerations**
> *Critique: "A more in-depth look at how these unique step sizes might affect adaptive step-size methods...would make the results more useful."*
>
> As we argue, the set of singular step-sizes has measure zero. This provides a strong guarantee of robustness for adaptive methods. Any adaptive algorithm or schedule that selects step-sizes from a continuous range (or even a sufficiently fine-grained discrete set) will have a zero probability of selecting a singular value. Therefore, our result implies that such methods do not need to be specially designed to avoid these specific pathological points, as they are naturally avoided.
>
> ### **On the Significance of Results**
> We thank the reviewer for this comment, which allows us to clarify our work’s novelty. To the best of our knowledge, existing proofs of non-singularity are limited to settings with strong assumptions. For example, the analysis in Lee et al. (2019) relies on Lipschitz smoothnesss, which, as we discuss on lines 97-98, does not hold for many practical deep learning scenarios. Our work is the first to provide a comprehensive proof of non-singularity for the GD map in the more general and practical setting of neural networks with piece-wise analytic activations. If there are specific references the reviewer believes we have overlooked, we would be very grateful for the pointers and would be happy to incorporate them into our discussion.
>
> ### **On the Novelty of the Proof Technique**
> Finally, we wish to gently counter the characterization of our proof as not being novel. As we detail in Remark 5, direct application of standard tools like the chain rule fail for the composition of almost everywhere analytic functions, which is the central challenge when moving beyond simple analytic activations. Our key technical innovation, presented in Proposition 6, is a novel "analogue of the chain rule" that leverages the specific layered structure of neural networks to overcome this limitation. To our knowledge, this inductive proof technique is a new contribution to the literature.

---

> > ### Comment · Reviewer_NUng · 2025-08-04
> >
> > Thank you for addressing my concerns

---

### Decision · Program_Chairs · 2025-09-17

**Decision:**

Accept (poster)

**Comment:**

The authors prove that for almost all learning rates the function that applies a gradient descent step to the parameters of a neural network never maps sets of positive measure to null sets, which has been assumed in many theoretical works but without a formal proof so far.

The paper is clearly written and features a novel proof technique.  It generalizes existing theoretical results to neural networks using popular activation functions such as ReLU, sigmoid, and leaky ReLU.  The results apply to a wide range of network architectures, such as fully connected, convolutional, and attention layers.

Reviewers expressed concerns about several issues, including setting biases to zero, correctness and completeness of handling convolutional and attention layers, connectedness of applications to the theoretical results, and relevance of the experimental results to practice.

The authors provided a comprehensive rebuttal, and there was some discussion following it.  The authors addressed most of the concerns, and several of the reviewers raised their ratings.  In the end, a consensus was reached among the reviewers, with all either proposing or not opposing acceptance.

I agree that this paper will provide an interesting contribution to the conference, with potentially significant impact on future theoretical works.  The problem is fundamental, and it was addressed successfully and in reasonable generality.